# Creating and validating the Fine-Grained Question Subjectivity Dataset (FQSD): A new benchmark for enhanced automatic subjective question answering systems

**Marzieh Babaali** [ID]**, Afsaneh Fatemi** [ID]*, **Mohammad Ali Nematbakhsh**

Faculty of Computer Engineering, University of Isfahan, Isfahan, Iran

* a_fatemi@eng.ui.ac.ir

**Data Availability Statement:** To ensure the reproducibility of our findings and facilitate further research in this area, the dataset used in this study and the corresponding code have been made

## Abstract

In the domain of question subjectivity classification, there exists a need for detailed datasets that can foster advancements in Automatic Subjective Question Answering (ASQA) systems. Addressing the prevailing research gaps, this paper introduces the Fine-Grained Question Subjectivity Dataset (FQSD), which comprises 10,000 questions. The dataset distinguishes between subjective and objective questions and offers additional categorizations such as Subjective-types (Target, Attitude, Reason, Yes/No, None) and Comparison-form (Single, Comparative). Annotation reliability was confirmed via robust evaluation techniques, yielding a Fleiss's Kappa score of 0.76 and Pearson correlation values up to 0.80 among three annotators. We benchmarked FQSD against existing datasets such as (Yu, Zha, and Chua 2012), SubjQA (Bjerva 2020), and ConvEx-DS (Hernandez-Bocanegra 2021). Our dataset excelled in scale, linguistic diversity, and syntactic complexity, establishing a new standard for future research. We employed visual methodologies to provide a nuanced understanding of the dataset and its classes. Utilizing transformer-based models like BERT, XLNET, and RoBERTa for validation, RoBERTa achieved an outstanding F1-score of 97%, confirming the dataset's efficacy for the advanced subjectivity classification task. Furthermore, we utilized Local Interpretable Model-agnostic Explanations (LIME) to elucidate model decision-making, ensuring transparent and reliable model predictions in subjectivity classification tasks.

## 1. Introduction

In the evolving landscape of e-commerce, platforms like Amazon have increasingly leveraged question answering (QA) systems to enhance customer experience. These systems tackle a spectrum of queries, ranging from subjective, grounded in personal beliefs and opinions [1–3], to objective, which are factual and unbiased [2, 3]. The emergence of Automatic Subjective Question Answering (ASQA) as a key subdomain within QA underscores the importance of delivering nuanced, opinion-based answers [4–6]. Central to ASQA's efficacy is its ability to

publicly available. The dataset can be accessed at https://github.com/mahsamb/FSQD, and the code repository is available at https://github.com/mahsamb/FSQD/blob/main/Figures.ipynb.

**Funding:** The author(s) received no specific funding for this work.

**Competing interests:** The authors have declared that no competing interests exist.

analyze and categorize questions accurately. While the distinction between subjective and objective questions is acknowledged, there exists a significant research gap in fine-grained subjectivity analysis. Current datasets, such as those by Yu et al. [7], SubjQA [8], and ConvEx-DS [9] are not exhaustive in their scope, hindering ASQA advancements.

Recognizing these gaps and building on the foundations laid by [7, 10], this study aims to provide a refined methodological approach towards question subjectivity. We focus particularly on smartphone-related questions and introduce an advanced fine-grained question subjectivity labeling scheme. This innovative framework offers 10 distinct labels, further elaborated through three sub-labels, clarifying the nuances of each question's subjectivity, comparison form, and subjective type. The design of this integrated system emphasizes the depth and multi-dimensional nature of our analysis, addressing the intricate objectives of this research. For a more tangible grasp, Table 1 provides exemplifications from the labeling scheme, elucidating every category and sub-category tailored for smartphone-centric queries.

The Subjective-type categorization serves to identify the specific nature of subjectivity within a given question [7, 10]. For example, "Target" refers to questions that solicit public opinion on a specific topic, while "Reason" includes questions that ask why such public opinions exist. The category "Attitude" encompasses questions that aim to capture the public's stance or feelings about a subject. "Yes/No" is designated for questions that either affirm or negate a particular statement about a topic. Lastly, the "None" type includes questions that are inherently objective.

The unified label scheme (fourth column, Table 1) offers a depth-focused analysis of each question's subjectivity, aligning with the research's granular objectives. Building on this labeling schema, we present the Fine-Grained Question Subjectivity Dataset (FQSD) comprising 10,000 questions. Rather than merely classifying questions as subjective or objective, FQSD further categorizes into Subjective-types (T, A, R, Y, N) and Comparison-form (C, S). A team of three annotators was employed, and rigorous evaluation techniques were applied to assess annotation consistency. The inter-annotator reliability was confirmed through strong statistical indicators: a Fleiss's Kappa score of 0.76 and Pearson correlation values reaching 0.80. These metrics Confirm both the reliability and validity of the FQSD dataset. The FQSD dataset was evaluated against existing datasets, including Yu et al. [7], SubjQA [8], and ConvEx-DS [9]. The findings indicate that FQSD excels in terms of scale, linguistic diversity, and syntactic complexity, setting a new benchmark for future research in Fine-grained Question Subjectivity Classification (FQSC). In addition to its superior metrics, TF-IDF were employed as a key visual methodology to provide a nuanced understanding of FQSD and its classes, further solidifying its role as a comprehensive resource for advancing the field of FQSC.

**Table 1. Samples of various question types.**

| Subjectivity | Subjective- Type | Comparison-Form | Question Label | Example |
|---|---|---|---|---|
| Subjective | Target | Single | **TSS:** Target Single Subjective | Which phone has a good camera with a price range of "120–210 $? |
| | | Comparative | **TCS:** Target Comparative Subjective | Which phone is slightly better than Samsung X? |
| | Attitude | Single | **ASS:** Attitude Single Subjective | What do people say about the sound quality of Samsung X? |
| | | Comparative | **ACS:** Attitude Comparative Subjective | What's better: iPhone X or Samsung Y? |
| | Reason | Single | **RSS:** Reason Single Subjective | Why do people recommend buying Samsung X? |
| | | Comparative | **RCS:** Reason Comparative Subjective | Why do people prefer iPhone X over Samsung Y? |
| | Yes/No | Single | **YSS:** Yes/No Single Subjective | Do people recommend buying Samsung X? |
| | | Comparative | **YCS:** Yes/No Comparative Subjective | Is Samsung as good as Samsung Y? |
| Objective | None | Single | **NSO:** None Single Objective | How long does the Samsung battery life last? |
| | | Comparative | **NCO:** None Comparative Objective | Which of Samsung X and Samsung Y are heavier? |

Following a comprehensive description of the FQSD dataset's characteristics, the research moved to evaluating its applicability through model experimentation. Transformer-based models (BERT, XLNET, and RoBERTa) were employed to determine their performance in classifying fine-grained question subjectivity. Among these, RoBERTa distinguished itself, achieving an impressive F1-score of 97%. In this exploration, Local Interpretable Model-agnostic Explanations (LIME) [11] were also implemented, providing a transparent viewpoint through which to perceive and understand the model's judgments on unseen data, thus ensuring the results are not only robust but also interpretable and reliable.

The contributions of this paper are as follows:

- The introduction of a balanced, multi-faceted FQSD, a 10,000-question dataset across 10 classes, focusing on smartphones.

- A unique four-category approach in the FQSD to improve subjectivity classification in QA systems.

- Emphasis on multi-sentence questions for real-world context.

- Annotation reliability confirmed with a Fleiss's Kappa of 0.76 and Pearson correlation up to 0.80.

- In-depth analysis and statistical visuals of FQSD's richness and complexity.

- Comparison of FQSD with three other datasets, highlighting its distinctive merits.

- RoBERTa's effectiveness in FQSC validated with an F1-score of 97%.

The rest of the paper is as follows: Section 2 discusses literature on question subjectivity classification. Section 3 introduces our task, dataset, and model for FQSC. Section 4 presents the evaluation of datasets and the proposed model. Section 5 concludes and suggests future research directions.

## 2. Related work

As the intricacies of Automated Question Answering Systems (AQA) are explored, understanding its foundational concepts and the advancements it has made over the years becomes vital. In Section 2, the importance of question analysis in AQA will be discussed, the development of question classification techniques from broad historical viewpoints to more detailed fine-grained perspectives will be outlined, and the inherent challenges in FQSC will be highlighted.

### 2.1 Question analysis in automated question answering systems

AQA systems have varied significantly across different domains, such as product reviews, social media interactions, customer service, and educational content. In this field, significant research has been conducted on question classification and its associated challenges. AQA systems primarily consist of three essential components: question analysis, information retrieval, and answer extraction [12–16].

Question analysis is a foundational phase in AQA, vital for enhancing its overall quality. Its main objective is to create a structured representation of the required information to address the user query [17–20]. Typically, this analysis involves two main tasks. The first, question processing, emphasizes identifying the question's core elements like keywords and target entities essential for information retrieval [14, 21]. The second task, question classification, categorizes

the question based on predefined types, which crucially impacts the answer extraction process and overall AQA system performance [22, 23].

## 2.2 Question classification: Historical to fine-grained perspectives

Within the field of Question-Answering (QA) research, question classification has established a significant path. Historically, researchers primarily explored factual questions, with seminal studies conducted by the likes of [24–26]. This empirical trend, however, began to diversify as the significance of subjective questions grew in prominence [4–6, 27].

The pioneering efforts of [27, 28] sparked this shift. Their individual utilization of SVM-based supervised models and semi-supervised co-training approaches established the groundwork for subjective question classification. A fascinating aspect of their methodologies was the dependence on feature extraction from both questions and corresponding answers. This, perhaps, poses a limitation, especially when compared with our research approach, which aims to classify questions leveraging inherent features without contingent answers. Further nuances in the subjectivity range emerged with [29] work, which, though achieving an 84.9% classification accuracy for Twitter-based questions, might have its generalizability queried given its platform-centric focus.

A detailed examination of the field highlights the progression from a basic objective-subjective differentiation, as formulated by [30], to more extensive and intricate classifications. [30] played a pioneering role by proposing a six-category classification for subjective questions. Subsequent scholarly work refined this framework. [31] introduced an 'Intensity' category, while researchers like [7, 10] adjusted these taxonomies to better suit product review contexts. Our investigation aligns with these innovative efforts but expands the focus to a three-fold categorization: subjectivity, subjective-type, and comparison-form.

While historically neglected, the domain of subjective comparative questions is now experiencing growing interest. Early research by [7, 10] often assigned comparisons to simple attributes, perhaps underestimating their classification potential. However, a pioneering study by Zhu et al. [6] renewed interest in this specialty, emphasizing the role of dependency relations in fine-tuning comparative patterns.

While the above-mentioned studies primarily focus on the product domain, recent research has extended into new areas. The creation and utilization of the CoQUAD datasets [32] for COVID-19 question-answering showcase a practical application of fine-grained question classification in a healthcare context. This work highlights the ability of ASQA systems to adapt and provide accurate, domain-specific information, addressing the complex challenges of classifying questions within specific domains like epidemiology, public health, and clinical care. This expansion reflects the evolving trends in ASQA research towards developing nuanced and effective question understanding mechanisms across diverse fields.

## 2.3 Challenges in FQSC

Building on insights from related work, FQSC, despite its rising significance, confronts a suite of challenges inhibiting its evolution:

- **Research Gaps:** While literature abounds with insights into subjectivity and comparison-form classifications, notably on a sentence level [33, 34], to the best of our knowledge, the confluence of these dimensions—subjective comparative questions—remains understudied. Bridging this lacuna is imperative for refining Automated Question-Answering Systems (ASQA).

- **Dataset Limitations:** The progression of research in this domain is stymied by a scarcity of manually curated datasets. Existing repositories such as Yu et al. [7] offer foundational, albeit confined, insights, incorporating solely single-sentence questions. Datasets like SubjQA [8] and ConvEx-DS [9] bring breadth to the table, but they misalign with the unique demands of our research, either by excluding comparative facets or subjective-type queries. Furthermore, the non-availability of datasets from pivotal studies exacerbates standardization woes, complicating benchmarking and comparative analyses.

- **Methodological Constraints:** Traditional methodologies, anchored in conventional machine learning and rule-based paradigms, present inherent challenges encompassing feature engineering, scalability, and adaptability [33].

In response to these identified challenges, the research undertakes a comprehensive examination of subjective comparative questions and investigates a diverse array of subjective subtypes. The FQSD—a carefully assembled dataset specifically designed for the domain of smartphones—is presented, serving as a foundation for the study on question subjectivity classification. Augmented by the utilization of transformer-based architectures, the methodological approach aims to overcome the limitations inherent in traditional methodologies. Consequently, a comprehensive and adaptable framework that addresses the prominent dataset and methodological common shortcomings in the domain is offered.

## 3. Methodology

In the methodology section, we embark on a comprehensive exploration of the foundational elements that underpin our study. Beginning with an in-depth introduction to the Fine-grained Question Subjectivity Dataset (FQSD), we detail the rigorous processes involved in dataset creation, ensuring high data quality and integrity. We then delve into the rationale behind selecting RoBERTa as our model of choice, emphasizing its architectural advantages and pre-training methodologies that are pivotal for accurately classifying nuanced subjective questions. Furthermore, we outline the model compilation and parameter selection strategies, highlighting the tailored optimization and evaluation techniques that mitigate overfitting and ensure robust model performance. This section lays the groundwork for understanding the meticulous approach and advanced techniques employed in advancing question subjectivity classification research.

### 3.1. Introduction to the FQSD

In this section, the intricacies of the FQSD, a meticulously curated collection designed to advance research in the nuanced field of question subjectivity classification, are explored. Comprising 10,000 carefully selected questions, the FQSD provides a balanced and comprehensive resource, facilitating a deeper understanding and development of NLP models tailored to discerning fine-grained subjectivity.

**3.1.1 Overview of the FQSD.** In this section, the proposed dataset, the FQSD, is described. This curated collection comprises 10,000 questions focused on various smartphones and their respective aspects. The dataset is systematically organized into 10 distinct classes, as specified in Table 2. Notably, each class is balanced, containing an equal distribution of 1,000 samples. This balanced composition ensures equitable representation across different question types, making FQSD an invaluable resource for research in FQSC.

**3.1.2 Descriptive metrics of the FQSD.** In Table 3, various descriptive metrics for the FQSD dataset are presented, highlighting its complexity and the richness of the information it contains.

**Table 2. Statistical information of FQSD dataset.**

| # | Task | Question Tags | FQSD Dataset |
|---|------|---------------|--------------|
| 1 | Subjectivity Classification | S | 8000 |
| | | O | 2000 |
| 2 | Subjective-Type Classification | A | 2000 |
| | | T | 2000 |
| | | R | 2000 |
| | | Y | 2000 |
| | | N | 2000 |
| 3 | Comparison-Form Classification | C | 5000 |
| | | S | 5000 |
| 4 | Subjectivity-ComparisonForm Classification | CS | 4000 |
| | | CO | 1000 |
| | | SS | 4000 |
| | | SO | 1000 |
| 5 | Fine-grained Question Subjectivity Classification (FQSC) | ACS | 1000 |
| | | ASS | 1000 |
| | | NCO | 1000 |
| | | NSO | 1000 |
| | | RCS | 1000 |
| | | RSS | 1000 |
| | | TCS | 1000 |
| | | TSS | 1000 |
| | | YCS | 1000 |
| | | YSS | 1000 |
| | **Total** | - | **10000** |

- **Total Questions:** The FQSD dataset contains a total of 10,000 questions, offering a robust sample size that is conducive for various types of analyses and machine learning models.

- **Total Words:** With 159,459 words, the dataset provides a substantial lexical diversity, making it suitable for understanding both the breadth and depth of the language constructs within the questions.

- **Unique Words:** The dataset features 4,736 unique words, underscoring the variety of terms used and suggesting a level of complexity in the questions.

**Table 3. Descriptive metrics for FQSD dataset.**

| Metric | Value |
|--------|-------|
| Total Questions | 10000 |
| Total Words | 159459 |
| Unique Words | 4736 |
| Average Words Per Sentence | 14.79 |
| Average Sentence Length (chars) | 81.79 |
| Average Word Length | 4.57 |
| Average Syllables Per Word | 1.60 |
| Multi-sentence Questions | 742 |

- **Average Words Per Sentence:** On average, each question contains approximately 14.79 words, indicating the questions are neither too brief nor excessively verbose.

- **Average Sentence Length (Characters):** The average length of a question in characters stands at about 81.79, further confirming that the questions are reasonably detailed.

- **Average Word Length:** The mean length of a word in the dataset is 4.57 characters, suggesting a balanced mix of simple and complex words.

- **Average Syllables Per Word:** The dataset has an average of 1.60 syllables per word, which provides insights into the phonetic complexity of the words used in the questions.

- **Multi-Sentence Questions:** A noteworthy feature is the inclusion of 742 multi-sentence questions. This is crucial for capturing the nuanced context of real-world queries and offers a more holistic understanding of how questions are posed, particularly on product question-answering platforms.

## 3.2. Data quality and integrity

Ensuring the standards of data quality and integrity is foundational to the development and utility of the FQSD dataset. This segment elaborates on the processes employed to cultivate a dataset that meets the demands of NLP research. Through comprehensive dataset preparation, the integration of unique features, a carefully designed annotation process, and stringent measures to gauge inter-annotator agreement, we establish an approach that prioritizes precision, reliability, and ethical considerations in dataset creation.

**3.2.1. Dataset preparation and specifications.** In this section, an exhaustive and detailed account of the rigorous preparation and underlying specifications that form the cornerstone of this dataset's development is provided. This includes an in-depth exploration of the methods employed, the criteria applied in the selection and adaptation of data, and the strategic considerations that guided the compilation process to ensure the dataset's integrity and relevance to its intended research applications.

- **Dataset Inspiration and Collection:** Our initial dataset drew inspiration from user-generated content on question-answering platforms such as Quora (www.quora.com) and Amazon (www.amazon.com), specifically in the smartphone domain. However, sourcing questions that met the specific criteria proved challenging. The criteria necessitated the explicit mention of compared entities' names, clarity, and specificity in phrasing, and alignment with our predefined categories.

- **Adaptation and Creation of the FQSD Dataset:** There is limited availability of questions directly fitting our research needs. To fill the gaps, we adapted existing content and created new questions, ensuring a rich variety of data. This included:

  - **Modify Sentences:** Transforming existing sentences into questions suitable for our study.

  - **Content Adjustment**: Adding, removing, or altering sections of the questions to enhance specificity and relevance.

  - **Creating Fictional Data**: Supplementing the dataset with fictional but plausible questions to cover a broader range of scenarios and inquiries.

- **Validation and Approval Process:** The altered and newly created questions were subjected to a rigorous validation process, overseen by expert commentators knowledgeable in writing

and conceptual aspects within the smartphone domain. Their approval ensured the dataset's relevance and adherence to our research objectives.

- **Bias Mitigation Strategies in Dataset Compilation:** To ensure the FQSD dataset minimizes biases inherent in question selection and framing, several strategies tailored to question classification tasks were implemented. Firstly, the dataset was compiled from a wide range of online platforms beyond mainstream sites like Quora and Amazon, including niche forums and industry-specific Q&A sites. This approach ensures a variety of question types and topics, reducing the risk of selection bias towards popular or commonly discussed issues. Secondly, during the adaptation and creation of questions, we carefully balanced the dataset to include questions from different smartphone categories, reflecting a broad spectrum of user intents and information needs. This diversity helps in minimizing content bias, ensuring our dataset supports the development of classification models capable of understanding a wide range of question formats and types. Finally, to address annotator bias in question classification, annotators were provided with extensive guidelines designed to maintain consistency and neutrality in question interpretation and classification. Regular calibration sessions were held to align annotators' understanding and ensure a uniform standard was applied across the dataset.

- **Compliance and Ethical Considerations:** Our data collection, modification, and creation processes were conducted in strict compliance with the terms and conditions of the source platforms. Ethical considerations were paramount, including the exclusion of any personally identifiable information and ensuring the privacy and confidentiality of individuals associated with the initial data sources.

**3.2.2. Unique features of the FQSD.** Notably, the FQSD dataset distinguishes itself by including multi-sentence questions, a crucial feature often overlooked but essential for capturing the nuanced context of real-world queries. Consider the following multi-sentence question: 'I have to buy a phone. I have only used Android phones, not tried Windows phones. My budget is under 25K. Are Windows phones better than Android phones?'. This question is dissected into four component sentences:

$S_1$: *'I have to buy a phone'* (Subjective, Non-comparative)

$S_2$: *'I have only used Android phones, not tried Windows phones'* (Objective, Non-comparative)

$S_3$: *'My budget is under 25K'* (Objective, Non-comparative)

$Q_1$: *'Are Windows phones better than Android phones*?' (Subjective, Comparative).

Here, $Q_1$ serves as the primary query, while $S_1$, $S_2$, and $S_3$ offer contextual information that might influence the interpretation of $Q_1$. By treating these sentences as a composite, multi-sentence question, our dataset offers a more holistic understanding of how questions are posed in real-world settings, particularly in product question-answering platforms.

In the meticulous construction of the FQSD dataset, our annotation process prioritized clarity and specificity by excluding questions that fall into multiple types, often arising from compound structures. This decision was guided by the observation that compound questions, such as 'Is A better than B and why?', inherently pose challenges for subjectivity classification by merging distinct inquiry types into a single query. By filtering out these questions, our dataset maintains a streamlined focus on single and straightforward multi-sentence questions, enhancing its utility for precise NLP tasks. This approach ensures the dataset's integrity by facilitating more consistent annotation and reducing the potential for bias, thereby solidifying the foundation for this research on subjectivity classification.

**3.2.3. Annotation process and inter-annotator agreement.** In our dedication to uphold the highest standards of data quality, the annotation process and inter-annotator agreement assessments were meticulously designed to ensure the integrity and reliability of the FQSD dataset.

We selected three PhD candidates with a focus on Natural Language Processing (NLP), each bringing a unique blend of expertise from various subfields within NLP such as computational linguistics, machine learning applications in language understanding, and text analytics. Their expertise was complemented by comprehensive training sessions focused on the nuances of FQSC, including workshops on recognizing and mitigating unconscious biases.

To guarantee annotation consistency, a majority vote system was employed for tag determination. Questions that elicited disparate tags were rigorously reviewed in follow-up sessions, allowing for consensus-building discussions that not only enhanced annotation quality but also served as an educational feedback loop for the annotators.

Reliable manual annotation is a critical factor for the validity and generalizability of any research derived from the dataset. To assess the quality and consistency of the annotations, the inter-annotator agreement was evaluated using both Pearson Correlation Coefficients and Fleiss's Kappa.

- **Pearson Correlation Coefficients:** Pearson correlation coefficients were calculated between each pair of annotators to quantify their level of agreement [35]. The coefficients range from -1, indicating perfect negative correlation, to +1, indicating perfect positive correlation. The calculated Pearson Correlation values for the annotator pairs are provided in Table 4:

The values in Table 4 suggest a substantial level of positive correlation, with the highest correlation observed between Annotator 1 and Annotator 3. This consistency enhances the overall reliability of the dataset.

- **Fleiss's Kappa:** Fleiss's Kappa is a robust statistical metric used to assess the reliability of agreement between multiple raters [36]. Unlike the Pearson correlation coefficient, Fleiss's Kappa accounts for the possibility of chance agreement, making it a more reliable measure. The computed Fleiss's Kappa value is 0.76. This value can be interpreted by referring to Table 5 below, which outlines different ranges of Fleiss's Kappa and their common

**Table 4. Pearson correlation among annotators.**

| Annotator Pair | Pearson Correlation |
|---|---|
| Annotator 1 & 2 | 0.67 |
| Annotator 2 & 3 | 0.73 |
| Annotator 1 & 3 | 0.80 |

**Table 5. Interpretation ranges for Fleiss's Kappa.**

| Fleiss's Kappa Range | Interpretation (Agreement Level) | Our Fleiss's Kappa |
|---|---|---|
| $< 0$ | Poor agreement | |
| 0.01–0.20 | Slight agreement | |
| 0.21–0.40 | Fair agreement | |
| 0.41–0.60 | Moderate agreement | |
| 0.61–0.80 | Substantial agreement | 0.76 |
| 0.81–1.0 | Almost perfect agreement | |

interpretations. The interpretation of the Kappa value is based on the ranges shown in Table 5, adapted from the work by [36].

According to Table 5, a Fleiss's Kappa value of 0.76 falls under the "Substantial agreement" category. This reinforces the dataset's reliability, affirming that it is a valuable asset for research in FQSC.

The inter-annotator agreement was quantified through Pearson Correlation Coefficients and Fleiss's Kappa, yielding values indicative of substantial agreement among the annotators. These measures reflect not only the consistency of the annotations but also the effectiveness of our bias mitigation strategies:

- Pearson Correlation showed significant positive agreement, with the highest alignment between Annotator 1 and Annotator 3, underscoring the rigorous training and selection process for our annotators.

- Fleiss's Kappa, standing at 0.76, further validated the reliability of our dataset, placing it within the 'Substantial agreement' range. This level of consistency is critical for the generalizability of research findings derived from the dataset.

By implementing these structured annotation processes and statistical validations, we aim to ensure that the FQSD dataset stands as a robust and unbiased tool for NLP research, facilitating advancements in the field with its high-quality, meticulously annotated questions.

## 3.3. Choice of model architecture

In the exploration of FQSC, RoBERTa emerged as the primary model of choice, a decision anchored in its exceptional architectural sophistication and pre-training advancements. Distinctively, RoBERTa's approach to pre-training—marked by an expansive dataset and iterative process augmented with dynamically changing masking patterns—catapults its linguistic comprehension to levels well-suited for the discerning differentiation between objective and subjective questions, alongside the precise categorization of question forms. The dynamic masking feature of RoBERTa signifies a critical innovation within the broader spectrum of context-aware language models, distinguishing it from other approaches in this category, such as static masking and permutation-based strategies. This feature is instrumental in broadening the model's contextual understanding, thereby amplifying its generalization capabilities to adeptly navigate the subtleties of unseen questions.

Crucially, RoBERTa abandons the Next-Sentence Prediction (NSP) task—a move that empirical evidence suggests significantly strengthens its efficacy in downstream applications, including the intricate domain of question classification. The model's deep Transformer layers, foundational to its architecture, serve as a robust feature extraction mechanism that can be particularly useful for the inherent complexities of question classification tasks. This architectural depth, combined with RoBERTa's proven adaptability across varied linguistic contexts and classification paradigms, underscores its optimal fit for our research goals. The model's demonstrated excellence across multiple NLP benchmarks further solidifies our confidence in its selection, ensuring our methodology is underpinned by the most advanced NLP technology available. Moreover, RoBERTa's capacity for granular text analysis is essential for delineating the ten subjectivity classes our study investigates. This careful selection of RoBERTa, based on superior performance and architectural merits, helps our study to be at the forefront of advancing FQSC.

Building on this foundation, the RoBERTa-base model, specifically chosen for our task, consists of 12 transformer layers, 768 hidden units per layer, and a total of 125 million

parameters. This architecture is designed to provide a deep and nuanced understanding of language, which is critical for the complex requirements of FQSC. By combining RoBERTa with LIME, we enhance not only the transparency of our analytical process but also enrich the understanding of the model's decision-making framework.

# 4. Experimental results

In this pivotal section, the outcomes of experimental endeavors are presented, spanning dataset evaluation, model assessment, and comparative analysis in the domain of Fine-grained Question Subjectivity Classification (FQSC). Beginning with a meticulous evaluation of datasets, informed by insights from TF-IDF analysis of the Fine-grained Question Subjectivity Dataset (FQSD), attributes, domains, and objectives are meticulously aligned to discern the optimal dataset for the study. Subsequently, a comprehensive evaluation of the RoBERTa model's performance in FQSC is embarked upon, employing a systematic approach that includes balanced datasets, fine-tuning strategies, and cross-validation techniques. Finally, a thorough comparative assessment of model performance across varied datasets is conducted, shedding light on the efficacy of the proposed model and its potential enhancements to the forefront of FQSC research. Through empirical evidence and rigorous analysis, critical insights essential for advancing research in Fine-grained Question Subjectivity Classification are unveiled.

## 4.1. Dataset evaluation

In this section, the evaluation of datasets is embarked upon, leveraging insights gained from the TF-IDF exploration of the Fine-grained Question Subjectivity Dataset (FQSD) discussed in the preceding subsection. Building upon the identification of key terms and language patterns, various datasets are scrutinized in the context of Fine-grained subjectivity classification research. By aligning attributes, domains, and objectives, the aim is to discern the optimal dataset for the study while emphasizing the unique contributions and utility of FQSD highlighted through TF-IDF analysis.

**4.1.1 TF-IDF's exploration of key terms within FQSD.** In this section, we leverage TF-IDF to visually dissect the FQSD, highlighting key terms in each class. TF-IDF, a statistical technique for determining the importance of a word in a document compared to a larger collection of documents, reveals significant and crucial terms, thus outlining unique language patterns within the various categories of the dataset. This method is instrumental in enhancing text analysis, allowing us to pinpoint and elevate the most telling words that define each category within FQSD.

Fig 1 presents the top 30 unigram, bigram, and trigram TF-IDF terms, emphasizing significant nouns and noun phrases within the FQSD dataset. Consistent with previous research [37, 38], entities and their attributes tend to manifest as nouns or noun phrases. The graph reveals prominent entities such as 'Samsung Galaxy', 'iPhone', and 'Xiaomi', alongside key aspects like 'camera', 'quality', and 'display', indicating these are focal points in questions. To achieve this, we utilize the spaCy library for precise natural language processing (NLP) to preprocess text, specifically targeting the extraction of significant nouns and noun phrases. The methodology employs the TfidfVectorizer from scikit-learn to compute TF-IDF scores, followed by sorting these scores to unearth the most impactful terms.

Furthermore, grasping the significance of adjectives and adverbs is essential for discerning subjectivity/objectivity and comparative/non-comparative context in textual data. In this context, for visual analysis via TF-IDF, we deliberately targeted these parts of speech within four streamlined classes: Comparative Subjective (CS), Comparative Objective (CO), Single Subjective (SS), and Single Objective (SO). This focus is justified by the observed linguistic overlap

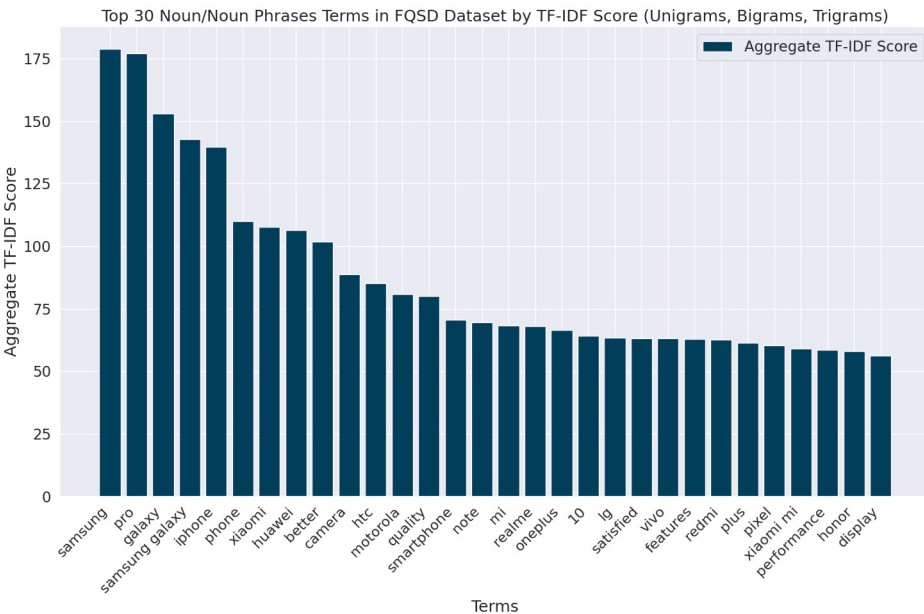

**Fig 1. Top 30 TF-IDF scores of nouns and noun phrases in FQSD.**

among the ten initial classes, where related categories demonstrated homogeneous usage of descriptors, thus obscuring the boundaries essential for detailed examination. By focusing the analysis on CS, CO, SS, and SO, analytical clarity is maintained, avoiding the dilution of distinctiveness and ensuring a concise representation of the data. In Fig 2, the top 20 terms for each class illuminate the discourse nature within these categories. For instance, "more," "higher," and "faster" are prominent in CO, suggesting quantifiable comparisons. CS terms like "better" and "superior" reflect evaluative judgments in comparative assessments. The SO class includes terms such as "fast," "compatible," and "expandable," focusing on objective attributes, while SS features terms like "satisfied," "well," and "good," indicating subjective personal assessments. To achieve this, we utilize the spaCy library to preprocess text, specifically targeting the extraction of significant adjectives and adverbs. The methodology employs the TfidfVectorizer from scikit-learn, configured to analyze only unigrams, to compute TF-IDF scores. These scores are then sorted to unearth the most impactful terms, emphasizing the focused and nuanced use of vocabulary within the FQSD dataset. It provides insights into the linguistic nuances and the focused use of vocabulary within the FQSD dataset.

In addition, Fig 3 showcases histograms of the TF-IDF scores for unigram adjectives and adverbs across four distinct classes—Comparative Objective (CO), Comparative Subjective (CS), Single Objective (SO), and Single Subjective (SS). It visualizes the distribution of non-zero TF-IDF scores for adjectives and adverbs in the dataset, and generates histograms to illustrate the variance and concentration of TF-IDF scores within each category. This visualization serves to further validate our analytical methodology. A closer examination of the histograms for each category uncovers the unique linguistic patterns that highlight our findings:

- **Comparative Objective (CO):** The histogram indicates a pronounced peak at higher TF-IDF scores, reflecting a concentrated use of specific terms critical for objective comparisons. Terms like "more," "higher," and "faster" dominate, suggesting a narrow, targeted linguistic approach in CO questions.

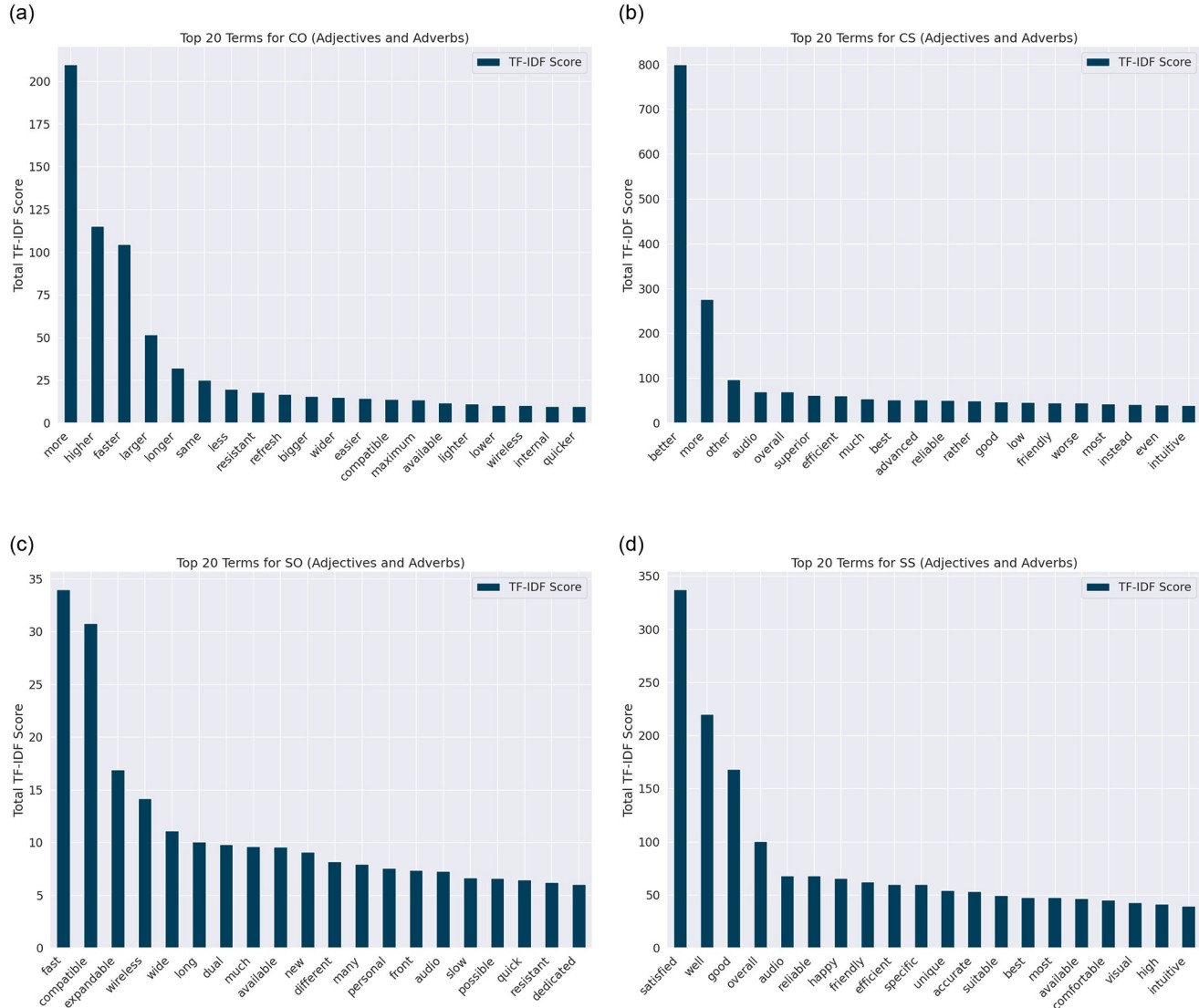

**Fig 2. Top 20 TF-IDF Scores of Adverbs/Adjectives in FQSD Categorized by Subjectivity Comparison-Form Classes: a) CO b) CS c) SO d) SS.**

- **Comparative Subjective (CS):** This class's histogram shows a broader distribution of TF-IDF scores, with a less sharp peak, indicating a diverse vocabulary used to express subjective comparisons. Terms such as "better" and "superior" are prevalent, highlighting the nuanced and varied language in CS questions.

- **Single Objective (SO) and Single Subjective (SS):** Both classes exhibit trends similar to their comparative counterparts, with SO showing a sharp peak and SS displaying a more even distribution of higher TF-IDF scores. The SO class focuses on specific objective attributes with terms like "fast," "compatible," and "expandable," while the SS class reveals a balanced use of language reflecting personal evaluations, highlighted by terms such as "satisfied," "well," and "good."

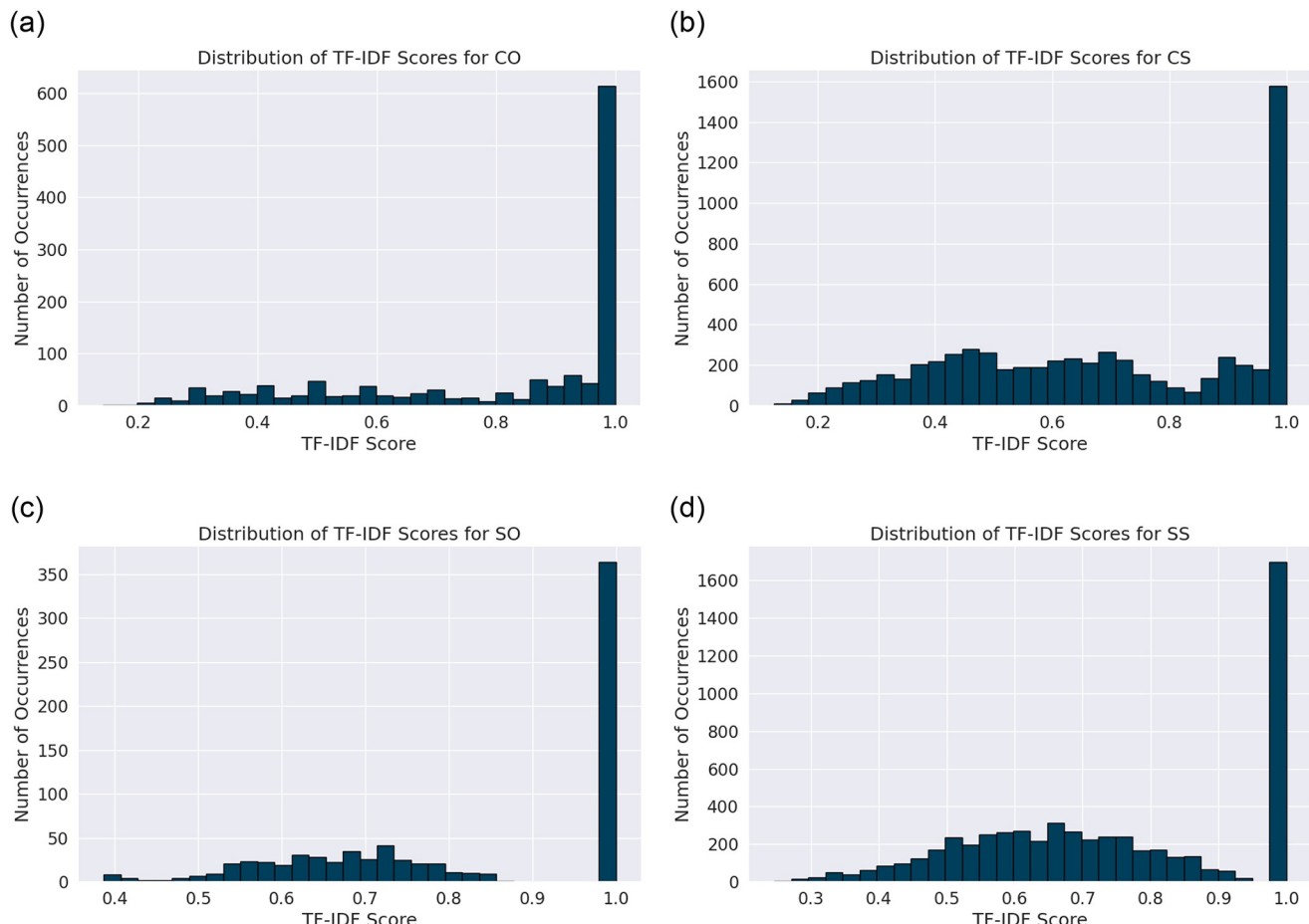

**Fig 3. Histograms of TF-IDF Adverbs/Adjectives Scores in FQSD Categorized by Subjectivity Comparison-Form Classes: a) CO b) CS c) SO d) SS.**

The distinct peaks at the higher end of the TF-IDF spectrum for CO and CS classes underscore the specificity of language in comparative questions, whereas the broader distribution in CS and the balanced distribution in SS suggest a richer vocabulary for subjective expressions and personal evaluations. These patterns validate the analytical approach, demonstrating that the four-class categorization effectively captures the linguistic essence of user queries in the dataset, showcasing clear lexical patterns that justify the segmented analysis for a nuanced understanding of language use within each category.

**4.1.2. Comparative analysis.** In this section, we undertake a comparative analysis to evaluate the FQSD dataset alongside three existing datasets commonly utilized in Fine-grained subjectivity classification research: the dataset of Yu et al. [7], ConvEx-DS [9], and SubjQA [8]. This analysis aims to provide insights into the unique strengths and comprehensive nature of FQSD, particularly in the context of FQSC. By examining various textual, syntactic, and semantic dimensions, we aim to highlight the distinct attributes and potential utility of FQSD as a gold-standard resource for researchers and practitioners in the field.

*Navigating datasets for question subjectivity.* In this section, different datasets currently utilized in the realm of various tasks of Fine-grained subjectivity classification research are explored. In the following, each dataset's attributes, domains, and how they align or diverge from the objectives of the study are detailed:

**Table 6. Statistical information of Yu et al. [7] dataset.**

| Task | Question Tags | Yu et al. [7] Dataset |
|---|---|---|
| Fine-grained Question Subjectivity Classification (FSQC) | ACS | 9 |
| | ASS | 36 |
| | NCO | 0 |
| | NSO | 55 |
| | RCS | 2 |
| | RSS | 43 |
| | TCS | 21 |
| | TSS | 1 |
| | YCS | 3 |
| | YSS | 50 |
| **Total** | - | **220** |

- **Dataset from Yu et al. [7]:** It is currently the only one closely related to our study, capable of covering all three tasks considered in this research. It comprises 220 questions from the cell phone domain (Refer to Table 6 for detailed statistics). It is a foundational resource but limited in its scope and complexity. It solely contains single-sentence questions, which may not be a true representation of intricate real-world queries.

- **ConvEx-DS Dataset [9]:** A newer dataset, comprising user question intents in the hotel domain. It contains 1,589 questions that cover both subjectivity and comparison-form classifications (Details in Table 7). It doesn't encompass subjective-type questions and not related to the smartphone domain.

- **SubjQA Dataset [8]:** The SubjQA dataset [8] offers a wealth of subjective and objective questions from six domains (Details in Table 8). While useful, it lacks features crucial to our study: it doesn't include comparison or subjective-type questions and isn't focused on the smartphone domain which is central to our research.

*Benchmarking against existing datasets.* In this section, a comparative analysis is undertaken to position the FQSD dataset against three existing datasets commonly used in the field: the dataset of Yu et al. [7], ConvEx-DS [9], and SubjQA [8]. The aim is to illuminate the unique strengths and comprehensive nature of FQSD, especially in the context of FQSC. The metrics are divided into two main categories: Textual Dimensions and Syntactic and Semantic

**Table 7. Statistical information of ConvEx-DS dataset [9].**

| Task | Classes | # Questions |
|---|---|---|
| Subjectivity | S | 915 |
| | O | 674 |
| Comparison-Form | C | 334 |
| | S | 1255 |
| Comparative Subjectivity | CS | 220 |
| | CO | 114 |
| | SS | 695 |
| | SO | 560 |
| | **Total** | **1589** |

**Table 8. Statistical information of SubjQA dataset [8].**

| Domain | Train | Dev | Test | Total |
|---|---|---|---|---|
| TripAdvisor | 1165 | 230 | 512 | **1686** |
| Restaurants | 1400 | 267 | 266 | **1683** |
| Movies | 1369 | 261 | 291 | **1677** |
| Books | 1314 | 256 | 345 | **1668** |
| Electronics | 1295 | 255 | 358 | **1659** |
| Grocery | 1124 | 218 | 591 | **1725** |

Dimensions. Each category contains various sub-metrics, offering a multi-faceted understanding of the datasets. By highlighting the distinctive attributes and sheer scale of FQSD, the potential of it as a gold-standard resource for researchers and practitioners alike is emphasized.

*Textual dimensions*. In this section, the dataset's textual characteristics are explored by analyzing its Size, Lexical Diversity, and Complexity. The dataset's Size and Structural Analysis are assessed through the number of questions it contains and the prevalence of multi-sentence questions, offering insight into its structural depth. Lexical Richness, including the Total and Unique Word Counts, reveal the dataset's vocabulary richness and linguistic variety. Furthermore, Lexical Complexity is examined through metrics like Average Words per Sentence, Average Sentence Length, Average Word Length, and Average Syllables per Word. These analyses, supported by specific metrics and visual aids, provide a comprehensive overview of the dataset's textual landscape, shedding light on its implications for complex language processing tasks and potential applications. This meticulous examination aims to understand the dataset's capacity to support advanced NLP challenges by highlighting its structural intricacies and linguistic diversity.

In the following sections, we will compare these metrics across four different datasets, offering a comparative analysis that highlights the unique characteristics, strengths, and limitations of each dataset concerning the others. This analysis, underpinned by specific metrics and visual aids, provides a detailed snapshot of each dataset's textual makeup and its potential implications.

*Size and structural analysis*. Understanding linguistic structures and vocabulary breadth is pivotal for NLP research, directly impacting model effectiveness. The analysis assesses both data volume and the syntactic intricacies within datasets to ensure they offer a diverse range of training examples and pose sufficient linguistic challenges for model refinement. By evaluating the Total Question Count for dataset size and using Multi-Sentence Question Count as a complexity metric—indicative of nuanced information structures—insights into each dataset's quantitative and qualitative dimensions are gained. This comprehensive approach informs dataset selection for NLP development, aiming to enhance models' linguistic understanding and generation capabilities.

To quantify these metrics, we initially segmented the text from each dataset into individual sentences using a regular expression. This process targeted common sentence-ending punctuation—specifically the terminal punctuation markers of '.', '!', and '?'—to delineate sentence boundaries. The Multi-Sentence Count metric was derived by algorithmically identifying sentences within each question based on terminal punctuation markers and calculating the proportion of questions containing more than one such marker.

Fig 4 presents a comparative analysis of datasets by the total number of questions (bar graph) and the count of multi-sentence questions (line graph). It reveals FQSD's dominance in

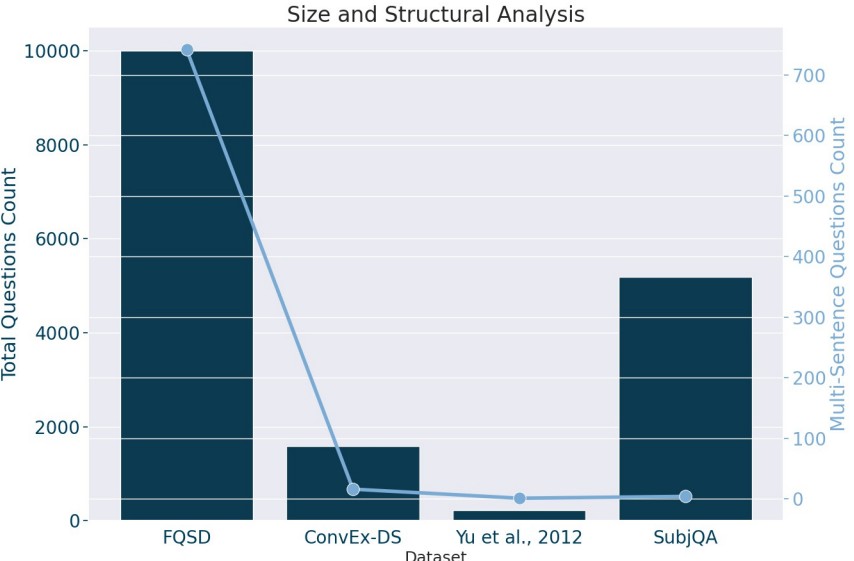

**Fig 4. Distribution of total question count and multi-sentence question count across the FQSD, ConvEx-DS, Yu et al., 2012, and SubjQA datasets, showcasing the size and structural analysis of each dataset.**

both Total Question Count and Multi-Sentence Question Count, marking it as the most extensive and structurally complex corpus among those studied. While SubjQA ranks second in question volume, its multi-sentence question count is much lower, indicating a simpler structure relative to its size. This compact visualization efficiently highlights the balance between dataset volume and complexity, crucial for selecting optimal corpora for NLP development, with FQSD offering a rich linguistic environment for comprehensive model training.

*Lexical richness.* In this investigation, we place a significant emphasis on the lexical composition of NLP datasets to determine their applicability for complex language tasks. Lexical metrics, such as the Total Word Count and Unique Word Count, offer insights into the textual richness and the diversity of the vocabulary. These metrics are crucial for evaluating the potential of datasets to provide varied linguistic input necessary for training robust NLP models [39–41]. By examining the total count of words alongside the count of unique words, the assessment not only evaluates the volume of linguistic content available but also its variety, which is indicative of the potential complexity and nuance that NLP models must grapple with.

To quantify these metrics, each dataset was first segmented into individual sentences. These sentences were then further refined by stripping extraneous whitespace and excluding empty sequences, ensuring a focus on substantive textual content. Subsequently, we applied Python's regular expressions for pattern matching to extract words, treating the boundary between word characters and non-word characters as the delimiter. This approach not only facilitated a comprehensive word count but also ensured that variations in case were normalized, thereby maintaining consistency in the unique word identification process.

Fig 5 reveals the outcomes of this lexical exploration, highlighting the FQSD dataset for its exceptional lexical breadth, as indicated by its high Total Word Count and significant count of unique words. This suggests not only a vast corpus but also a high degree of vocabulary diversity, making it an ideal resource for NLP tasks requiring extensive linguistic variation. In contrast, SubjQA, while smaller in total word volume, also exhibits a noteworthy proportion of

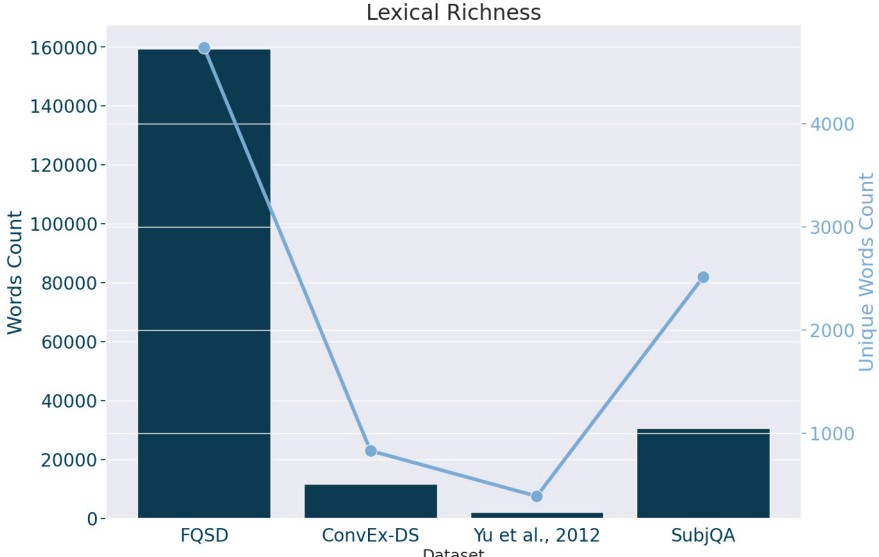

**Fig 5. Distribution of the total word count and unique word count across the FQSD, ConvEx-DS, Yu et al., 2012, and SubjQA datasets, showcasing the lexical richness of each dataset.**

unique terms, underscoring its value in providing varied linguistic inputs. This detailed comparison, as presented in Fig 5, serves as a critical resource for selecting datasets that offer the lexical depth necessary for advanced NLP model development and evaluation.

*Lexical complexity*. In this linguistic analysis, we sought to capture the textual complexity of various datasets by evaluating both the structural and lexical properties of the text. This encompassed a multifaceted approach where we examined Average Words per Sentence [42, 43], Average Sentence Length [42, 44], Average Word Length [45, 46], and Average Syllable Count per Word [42, 46, 47]. These metrics collectively paint a picture of the textual density and readability, which are indicative of the cognitive load required for comprehension and have implications for the complexity of NLP tasks.

We used the re-library for regular expressions to segment texts into sentences and words. A specialized function was implemented to assess phonetic complexity, systematically estimating the syllable count of each word. This function employs a rule-based algorithm that processes words in lowercase for consistency and uses heuristic methods to tally syllables. It accounts for the phonetic rules of English, such as incrementing the syllable count when encountering vowels at the beginning of a word or following a consonant, and decrementing for silent terminal 'e's, with exceptions like 'le' endings. A final validation step ensures that every word is attributed at least one syllable, adhering to the fundamental structure of language. This meticulous syllable estimation enriches our textual complexity analysis, providing an integral measure that, despite its algorithmic simplicity, yields a robust gauge of the phonetic dimension within the corpora.

Fig 6 presents the results of this analysis in a visually accessible format, with each bar representing a different aspect of textual complexity across the datasets. The FQSD dataset distinctly outperformed in all four assessed metrics: Average Words per Sentence, Average Sentence Length, Average Word Length, and Average Syllables per Word. This comprehensive lead in lexical metrics underscores FQSD's extensive linguistic structure and phonetic complexity. Following in second place, the 'Yu et al., 2012' dataset showed commendable lexical density and complexity, albeit not matching the breadth of FQSD. The 'ConvEx-DS' dataset occupied

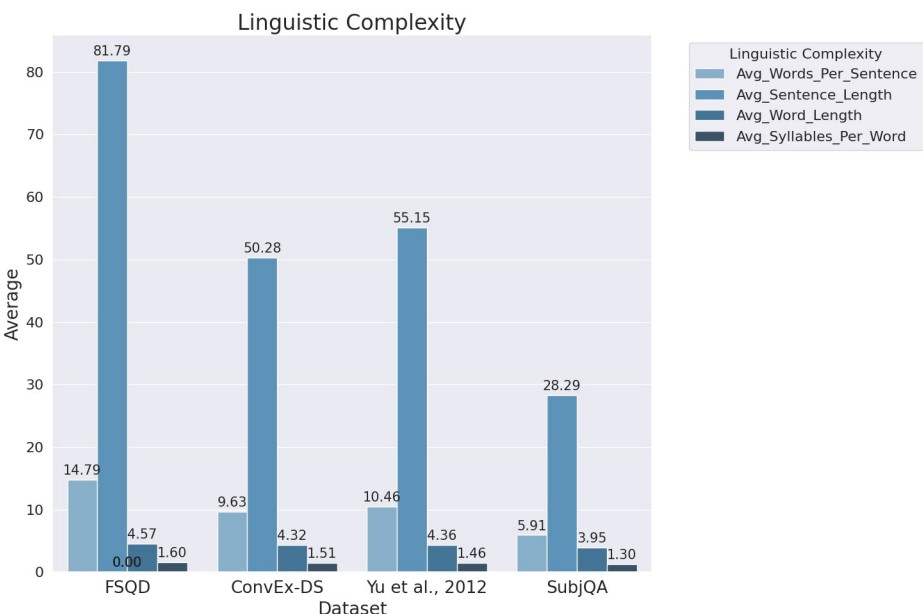

**Fig 6. Distribution of the average words per sentence, average sentence length, average word length, and average syllables per word across the FQSD, ConvEx-DS, Yu et al., 2012, and SubjQA datasets, showcasing the linguistic complexity of each dataset.**

the third position, indicating a moderate level of textual intricacy. In contrast, 'SubjQA' registered the lowest values across these metrics, suggesting a relative simplicity in its textual composition when compared to the others. This hierarchy of datasets in terms of textual complexity provides pivotal insights for NLP research.

*Syntactic and semantic dimensions*. In the "Syntactic and Semantic Dimensions" section, we scrutinize the datasets' structural and linguistic depth using metrics such as Average Parse Tree Depth, Mean Dependency Distance, Root Type-Token Ratio (RTTR), Corrected Type-Token Ratio (CTTR), and Sparsity Degree, complemented by visual references. This exploration reveals the Syntactic Complexity, showcasing the intricacies of sentence structures; Dependency Analysis, highlighting word interconnections; and Lexical Diversity, measuring vocabulary richness. Additionally, we assess Data Sparsity to understand the presence of rare elements, crucial for modeling challenges in NLP tasks. These analyses collectively illuminate the datasets' capabilities and limitations, providing a nuanced understanding of their syntactic and semantic attributes essential for advanced linguistic processing and application development.

*Syntactic complexity*. Syntactic Complexity refers to the hierarchical structure and arrangement of elements within sentences, serving as a key indicator of the cognitive and linguistic challenges a text presents. In this study, a detailed approach is employed to quantify this complexity through the analysis of Average Parse Tree Depths across four datasets, highlighted in Fig 7. The role of nouns and noun phrases is central to this measure, as they often form the core constituents of the sentence structure, serving as subjects, objects, or complements. The depth of the parse tree reflects the level of nestedness and the complexity of these syntactic constructions. Trees with deeper structures often contain noun phrases with additional modifying elements, such as adjectives and prepositional phrases, which contribute to the intricacy of the language used in the text.

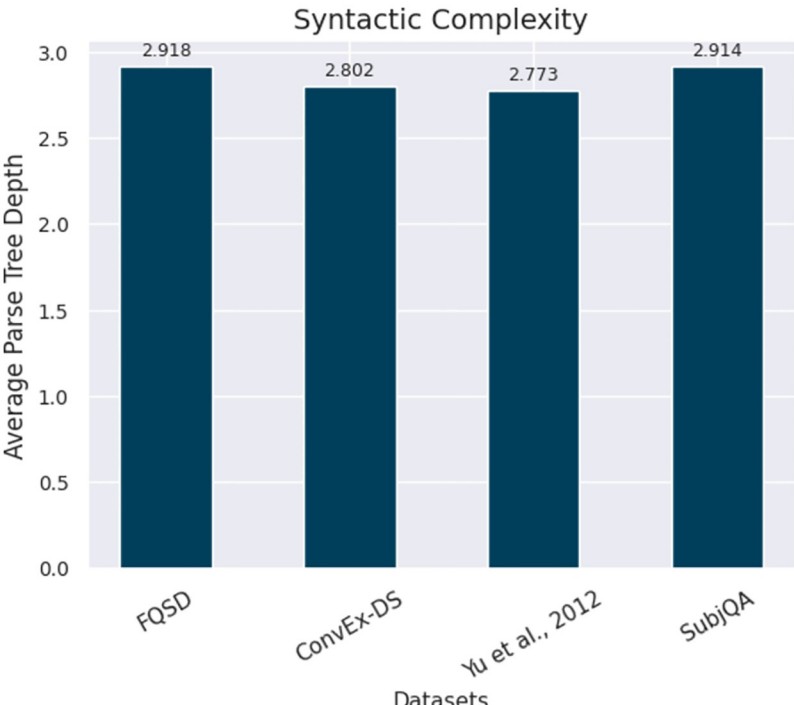

**Fig 7. Visualizing the average parse tree depth across the FQSD, ConvEx-DS, Yu et al., 2012, and SubjQA datasets, showcasing the syntactic complexity of each dataset.**

The Average Parse Tree Depths—a result of calculating the longest path from the root to any leaf within the tree—serves as an aggregate indicator of the dataset's syntactic complexity [48–50]. This metric is particularly insightful for understanding the variations and intricacies in the datasets, crucial for the development of robust NLP systems. The presence of complex noun phrases within these trees indicates the potential cognitive load required to process and understand the text, which is a vital consideration in NLP applications like machine translation and information extraction.

The analysis revealed in Fig 7 shows that the FQSD dataset exhibits the highest level of syntactic complexity, implying a dense layering of linguistic constructs within its parse trees. In contrast, the Yu et al., 2012 dataset presents a lower complexity level, suggesting more direct and potentially simpler syntactic structures. The ability to discern these differences is essential when curating corpora for NLP tasks, especially in scenarios where the depth of linguistic analysis, such as identifying subjective nuances within text, is critical.

*Dependency analysis.* In this study, a method is utilized to quantify syntactic complexity across various text datasets by calculating the Mean Dependency Distance (MDD), a metric indicative of linguistic intricacy [51–53]. Utilizing the spaCy library, the MDD metric gauges the average number of words that separate a word from its syntactic head within a sentence. Specifically, the MDD is derived by parsing each sentence to identify these dependencies, calculating the absolute differences between the indices of each word and its head, and then averaging these distances across the entire text. Higher MDD values indicate sentences with greater complexity, where words are, on average, further apart from their syntactic heads, while lower values suggest a tighter, more direct sentence structure.

Fig 8 illustrates the MDD values for four distinct datasets, derived from the application of a spaCy-based NLP model. The FQSD dataset exhibits the highest MDD, signifying its status as the most syntactically complex corpus among those studied. This complexity could pose more challenges but also offers potentially more opportunities for NLP models to learn nuanced language patterns. In contrast, the lower MDD of SubjQA suggests simpler syntax, potentially facilitating comprehension by NLP models. The ConvEx-DS and Yu et al., 2012 datasets have moderate MDD values, denoting a balanced syntactic structure. The differences in MDD demonstrated in Fig 8 highlight the variability in syntactic construction across datasets and emphasize the importance of considering syntactic complexity in NLP model development and dataset selection.

*Lexical diversity.* The metric of lexical diversity, determining the spread of unique words, provides a window into the text's varied linguistic fabric and complexity. The metric of lexical diversity, determining the spread of unique words, provides a window into the text's varied linguistic fabric and complexity [54]. In this study, we utilize the Root Type-Token Ratio (RTTR) and the Corrected Type-Token Ratio (CTTR), leveraging the spaCy NLP library, to analyze and quantify this aspect of textual data. These metrics are essential for evaluating the richness of a dataset's vocabulary, which is an important factor in various NLP applications.

The RTTR [55] is a normalization of the Type-Token Ratio (TTR) [55], which is the number of unique words (types) divided by the total number of words (tokens) in a text. Since TTR is affected by the length of the text, RTTR is used to standardize this measure across texts of different lengths by dividing the number of unique words by the square root of the total number of words, thus providing a more consistent metric of lexical diversity regardless of text size.

The CTTR [55] further corrects for the influence of text length on TTR. It divides the number of unique words by the square root of twice the total number of words, offering a more

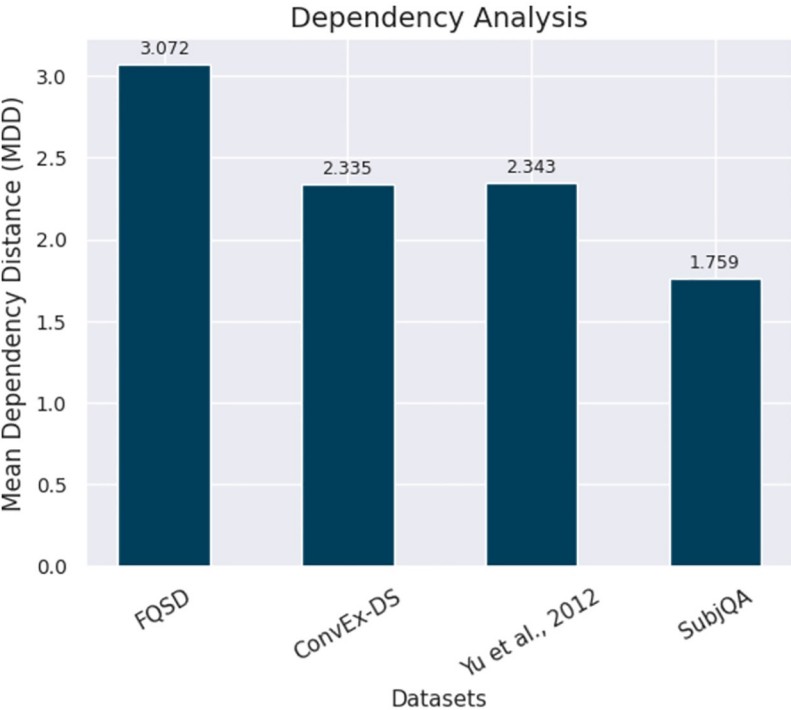

**Fig 8. Visualizing the Mean Dependency Distance (MDD) across the FQSD, ConvEx-DS, Yu et al., 2012, and SubjQA datasets, showcasing the dependency analysis of each dataset.**

balanced measure of lexical richness, particularly in longer texts, where the sheer volume can skew the traditional TTR.

Fig 9 assesses the lexical diversity of four datasets through the RTTR and the CTTR, with RTTR values depicted in dark blue and CTTR in light blue. The FQSD dataset is distinguished by the highest RTTR and CTTR values, indicating substantial lexical diversity suitable for robust NLP training. In contrast, Yu et al., 2012 exhibit the least diversity, with the lowest RTTR and CTTR values, pointing to a more limited or focused lexical scope. SubjQA and ConvEx-DS display moderate diversity, suggesting a balanced range of vocabulary. This visualization underscores the varying lexical complexities across datasets, which is essential for selecting appropriate corpora for specific NLP tasks.

*Data sparsity*. In the pursuit of enhancing our understanding and analysis of datasets used for FQSC, we adopted a metric for assessing the linguistic diversity of each dataset: Sparsity Degree. This metric is pivotal in quantifying the extent of vocabulary diversity, which is essential for tasks requiring nuanced language interpretation, such as subjectivity classification. The Sparsity Degree is calculated using the formula [56]:

$$Sd = 1 - \frac{1}{n} \sum_{i=1}^{n} \frac{N_i}{|V|} \qquad (1)$$

Where *Sd* represents the Sparsity Degree, *n* is the total number of questions in the dataset, $N_i$ denotes the number of distinct words in the *i*-th question, and $|V|$ signifies the size of the vocabulary, which is the total number of unique words across the entire dataset.

This formula provides a nuanced measure of sparsity by not merely counting unique terms but by evaluating their distribution across the dataset. A higher Sparsity Degree indicates a dataset with a broader lexicon and less repetition of individual words, suggesting a rich linguistic diversity. Such diversity is instrumental for developing models capable of understanding and classifying the subtle gradations of subjectivity expressed in questions, as it exposes the model to a wide variety of linguistic constructs and semantic nuances.

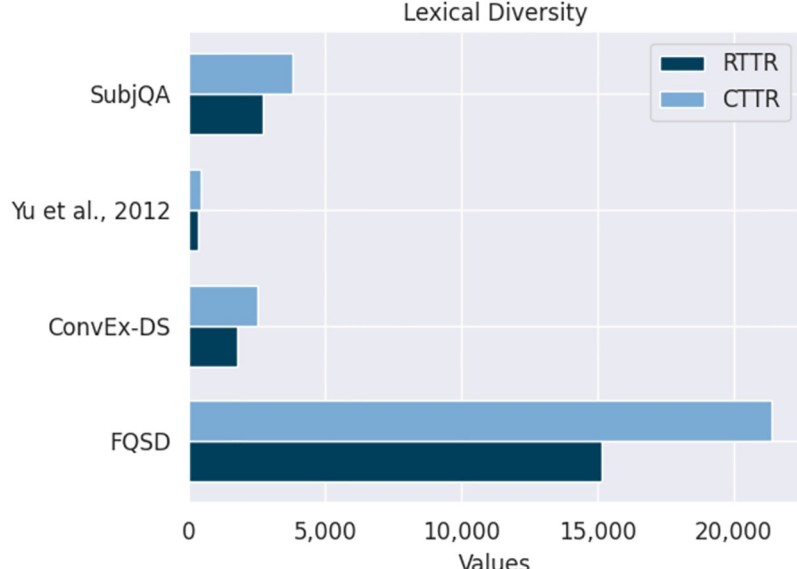

**Fig 9. Visualizing the Root Type-Token Ratio (RTTR) and the Corrected Type-Token Ratio (CTTR) across the FQSD, ConvEx-DS, Yu et al., 2012, and SubjQA datasets, showcasing the lexical diversity of each dataset.**

When evaluating Sparsity Degree within datasets, it's essential to also consider the number of questions, as high sparsity in larger datasets indicates a challenging achievement of vocabulary diversity, reflecting content richness. This aspect significantly impacts model training, where larger, diverse datasets facilitate learning and generalization, unlike smaller ones where high sparsity may not equate to lexical richness. Thus, assessing both Sparsity Degree and dataset size provides a nuanced perspective, ensuring interpretations are well-founded and reflective of the dataset's true linguistic complexity.

Considering both the Sparsity Degree values and the Question Counts for each dataset (Fig 10), we can derive a more nuanced interpretation of their characteristics and potential value for NLP tasks:

- **FQSD**: With a Sparsity Degree of 0.987 and 10,000 questions, FQSD stands out not only for its exceptionally high vocabulary diversity but also for its substantial size. The combination of a high Sparsity Degree with a large number of questions suggests that FQSD is incredibly rich in unique content, covering a wide range of topics and expressions. This makes it an ideal dataset for developing and testing NLP models, especially for tasks requiring nuanced language understanding, such as FQSC.

- **ConvEx-DS**: Despite having a smaller size of 1,589 questions, ConvEx-DS has a Sparsity Degree of 0.985, indicating a high level of vocabulary diversity. The smaller size combined with high diversity suggests that ConvEx-DS is focused yet varied, potentially offering concentrated examples of specific linguistic phenomena or domain-specific language that could be valuable for targeted NLP applications.

- **Yu et al., 2012**: This dataset is the smallest with only 220 questions, yet it maintains a high sparsity degree of 0.981. The high sparsity degree here could be influenced by the dataset's small size, meaning each question potentially introduces unique terms not seen elsewhere in the dataset. The small size and high sparsity degree could pose challenges for model training due to the limited amount of data.

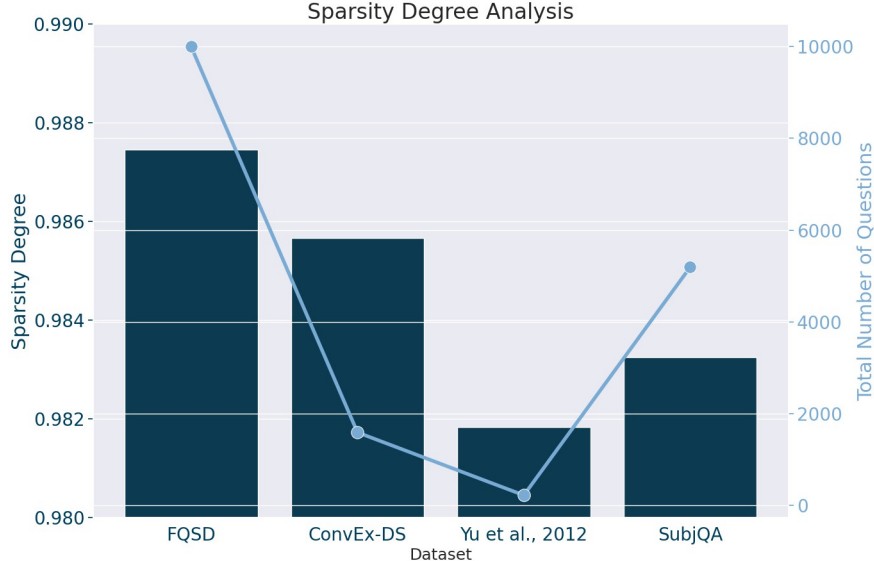

**Fig 10. Visualizing the sparsity degree and total question count across the FQSD, ConvEx-DS, Yu et al., 2012, and SubjQA datasets, showcasing the data sparsity of each dataset.**

- **SubjQA**: With 5,192 questions and a Sparsity Degree of 0.983, SubjQA offers a good balance between size and vocabulary diversity. This suggests that SubjQA provides a substantial amount of varied content, making it suitable for training models that need to generalize across a range of topics and styles. Its relatively high Sparsity Degree, combined with a moderate question count, indicates that it is both broad and rich in unique linguistic content.

In summary, FQSD, with its extensive size and the highest sparsity degree, stands out as an ideal dataset for training robust NLP models equipped to handle a broad spectrum of expressions and topics.

*Summary and implications*. The FQSD dataset stands out in the field of FQSC with its significant scale and diversity. With 10,000 questions and 159,459 total words, it sets a high standard for dataset size and complexity. The dataset's richness is further emphasized by its lexical diversity, consisting of 4,736 unique words. This not only makes it a substantial resource for various NLP tasks but also suggests good generalizability for models trained on it.

In terms of lexical and syntactic metrics, FQSD offers intricate complexities, including an Average Parse Tree Depth of 2.924 and a mean dependency distance of 2.8837, which make it a valuable resource for advanced linguistic studies and machine learning applications. Its lexical diversity metrics, RTTR and CTTR, are notably high, indicating linguistic richness. The low Sparsity Degree further adds to its robustness and utility.

Overall, the FQSD dataset serves as an invaluable tool for the community, offering a unique blend of size, complexity, and linguistic diversity. It is well-suited for advancing the field of FQSC and should be considered as a benchmark for future research.

## 4.2 Model evaluation

In this section, we embark on a comprehensive evaluation of the RoBERTa model's performance in the context of Fine-Grained Question Subjectivity Classification (FQSC). Through a meticulous analysis employing a balanced dataset, fine-tuning strategies, and cross-validation, we aim to ascertain the model's efficacy and reliability across diverse subjectivity classes. This systematic approach allows us to not only validate the model's predictive accuracy but also to explore the nuances of its decision-making process and the impact of dataset size on its performance.

**4.2.1 Evaluating RoBERTa's performance in FQSC.** The task of FQSC is addressed by employing the RoBERTa model. A balanced dataset consisting of 10 different classes that reflect varying degrees of subjectivity is curated. The data is randomly split into 5 different folds to perform 5-fold cross-validation. For model fine-tuning, the Adam optimizer was used, testing learning rates from {1e-5, 2e-5, 3e-5}, batch sizes of {16, 32, 64}, and epochs from {2, 3, 4, 5}. Based on validation performance, a learning rate of 1e-5, batch size of 32, and 5 epochs were selected for the experiments. The evaluation results are summarized in the Table 9. To

**Table 9. RoBERTa's five-fold cross-validation evaluation on FQSD.**

| Fold | Precision (%) | Recall (%) | F1-Score (%) |
|---|---|---|---|
| 1 | 96.8 | 96.6 | 96.6 |
| 2 | 96.0 | 95.9 | 95.8 |
| 3 | 96.9 | 96.8 | 96.8 |
| 4 | 97.7 | 97.7 | 97.6 |
| 5 | 97.8 | 97.8 | 97.8 |
| **Average** | **97.04** | **97.16** | **97.12** |

mitigate overfitting, dropout layers with a rate of 0.5 were incorporated, and L1 regularization was applied to reduce reliance on individual neurons and improve feature generalization across different data points. The evaluation results are concisely summarized in Table 9.

Across its five-fold cross-validation, RoBERTa exhibited remarkable consistency and robustness, with mean Precision, Recall, and F1-Score nearing 97%, based on five independent runs. This performance reinforces the model's accuracy and reliability in FQSC, showcasing its capability to handle the complexity and variability inherent in the FQSD dataset. These results, attesting to RoBERTa's potential for real-world applications in fine-grained subjectivity analysis.

**4.2.2 LIME visualizations: Unveiling decision dynamics on unseen data.** To elucidate the decision-making processes of the model and ensure the interpretability of its predictions, Local Interpretable Model-agnostic Explanations (LIME) was employed. LIME provides a transparent view into the model's reasoning, highlighting the words and phrases that significantly influence its predictions. In Fig 11, LIME visualizations of word influence in the model's predictions for Instances 1 and 2 on the Yu et al. [7] dataset are effectively illustrated.

On the left side of each figure, the prediction probabilities for different classes are displayed. "Other" represents the cumulative probability of six additional classes due to space limitations. Despite "Other" showing a higher cumulative probability, the individual probabilities for "TSS" in Fig 11(a) and "YSS" in Fig 11(b) stand out, marking them as the model's top predictions for the respective instances. Specifically, in sub-figure (a), although "Other" encapsulates a cumulative probability of 0.51 for RCS, ACS, YCS, YSS, and NCO classes, "TSS" demonstrates the highest individual class probability at 0.23. This makes "TSS" the model's leading prediction for the question "What features of iPhone 3gs are disliked by reviewers?

The right side of each figure visualizes the word-level contributions to the model's prediction. Words that positively influence the prediction towards "TSS" or "YSS" extend the bars to the right. Conversely, words that negatively influence these predictions extend to the left. For instance, in Fig 11(a), the LIME analysis reveals that words such as "What" and "disliked" carry significant weight, with "disliked" being especially impactful, suggesting a strong subjective sentiment. Similarly, in Fig 11(b), words like "fragile" and "broken" significantly influence the model's classification of the sentence as "YSS", indicating subjectivity based on personal experience or opinion.

The section titled "Text "ith high"ighted words" at the bottom presents the actual questions with key influential words highlighted. This provides a direct correlation to the visualized word weights and allows for an intuitive understanding of the model's decision-making process.

Overall, these LIME visualizations are instrumental in dissecting the RoBERTa model's reasoning on a per-word basis. They offer deep insights into the model's logic, ensuring that the predictions are based on relevant textual evidence.

**4.2.3 Influence of dataset size on model performance.** To assess the impact of dataset size on model performance, the data was analyzed at specific proportions relevant both academically and practically. This approach highlights performance variations with changes in dataset size. The analysis considered the full dataset (10,000 samples), three-quarters (7,500 samples), half (5,000 samples), one-quarter (2,500 samples), and one-tenth (1,000 samples). Furthermore, to ensure fairness and consistency in the evaluation, a stratified 5-fold cross-validation approach was utilized. This validation method ensures that each fold maintains the same distribution of classes.

Key Observations from Table 10:

(a)

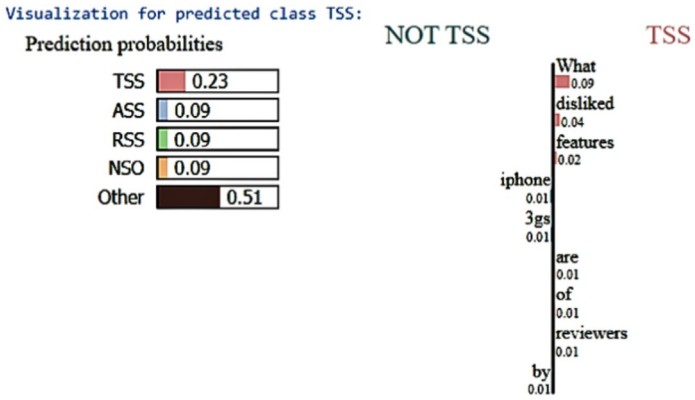

(b)

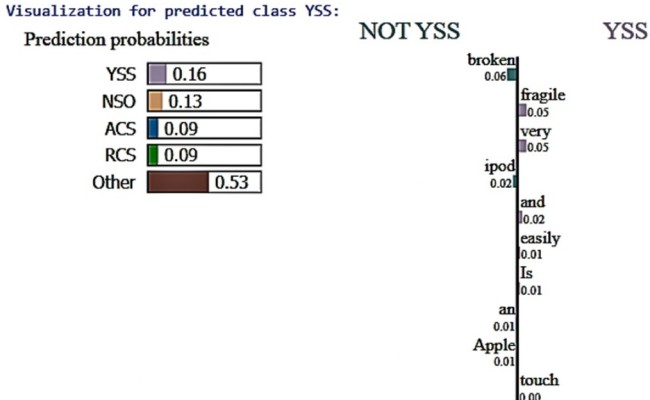

**Fig 11. LIME visualizations of word influence in model's predictions for Instances 1 (Fig 11a), and 2 (Fig 11b) on the Yu et al. [7] dataset.**

**Table 10. Model performance across different dataset sizes (averaged over 5 runs using stratified 5-fold cross-validation).**

| Dataset Portion | Dataset Size | F1-Score (%) | STD (%) |
| --- | --- | --- | --- |
| Full Dataset | 10,000 | 97.88 | 0.16 |
| Three-Quarters | 7,500 | 97.00 | 0.51 |
| Half | 5,000 | 96.11 | 0.74 |
| One-Quarter | 2,500 | 95.56 | 1.09 |
| One-Tenth | 1,000 | 78.88 | 6.21 |

- The model exhibits its zenith performance with the Full Dataset, achieving an F1-Score of 97.88% with a minimal STD of 0.16%.

- As we transition through the milestones, a discernible decline in the F1-Score is evident, with the most significant dip occurring at the One-Tenth milestone.

- The escalating STD as the dataset size decreases suggests increasing variability in the model's performance.

The systematic milestone-based approach clearly underscores the profound impact of dataset size on performance. The sharp drop in performance at the One-Tenth milestone and the accompanying rise in variability emphasize the importance of data volume and quality. Coupled with stratified 5-fold cross-validation, our methodology offers robust and insightful findings on the dataset-model performance relationship.

## 4.3. Comparative assessment

In this section, we delve into a thorough comparative assessment of model performance across varied datasets, highlighting the efficacy of the proposed model in Fine-grained Question Subjectivity Classification (FQSC). Through cross-dataset evaluations on SUBJQA [8] and ConvEx-DS [9] datasets, the model's adaptability to diverse linguistic contexts and domain-specific nuances is scrutinized. Moreover, a tailored evaluation strategy on the Yu et al. [7] dataset underscores the model's generalizability and effectiveness in real-world applications. Subsequently, a juxtaposition of the model with the Yu et al. [7] model across multiple classification paradigms showcases incremental improvements in FQSC. This comprehensive analysis illuminates the robustness and potential enhancements the model brings to the forefront of FQSC research.

**4.3.1. Cross-dataset performance evaluation of proposed model.** The efficacy of any model in NLP is best measured by its performance across varied and independent datasets. Such cross-dataset evaluations not only challenge the model's ability to generalize but also reveal its potential to adapt to different linguistic contexts and domain-specific nuances. Within this section, a comprehensive examination of the proposed model's capabilities is undertaken, reflected through rigorous testing on multiple datasets.

Tables 11 and 12 meticulously enumerate the analytic results of the proposed model, evaluated over SUBJQA [8] and ConvEx-DS [9] datasets, revealing the model's adeptness across diverse domains and through varied k-fold validations, thereby illuminating its robustness and adaptability to assorted data characteristics.

**Table 11. Analysis of the proposed subjectivity classification model over five separate runs on the SUBJQA dataset [8].**

| Domain | Precision | Recall | F-Score |
|---|---|---|---|
| Books | 89.70 | 88.03 | 88.57 |
| Electronic | 91.72 | 91.59 | 91.65 |
| Grocery | 87.12 | 86.04 | 86.44 |
| Movies | 94.22 | 94.17 | 94.20 |
| TripAdvisor | 94.40 | 93.86 | 94.04 |
| Restaurants | 93.15 | 93.25 | 93.15 |
| Average | **91.72** | **91.32** | **91.18** |

**Table 12. Analysis of the proposed subjectivity-comparison form classification model over five separate runs on the ConvEx-DS dataset [9].**

| K-fold | Precision | Recall | F-Score |
|---|---|---|---|
| 1 | 89.23 | 90.10 | 89.66 |
| 2 | 91.42 | 89.25 | 90.32 |
| 3 | 90.33 | 89.37 | 89.85 |
| 4 | 90.44 | 91.26 | 90.84 |
| 5 | 91.18 | 90.21 | 90.69 |
| Average | **90.52** | **90.04** | **90.07** |

In light of the limited volume of the Yu et al. [7] dataset, which comprises a modest total of 220 questions, a tailored evaluation strategy was imperative. The proposed model was trained on the comprehensive FQSD dataset to ensure it benefited from a broad spectrum of subjectivity nuances. After the training phase, we leveraged the Yu et al. [7] dataset as a test set. This approach allowed us to evaluate the model's performance on an independent dataset and to affirm its generalizability and effectiveness in a cross-dataset application.

Building upon this foundation, Table 14 juxtaposes our model with the Yu et al. [7] model across four classification paradigms: Subjectivity, Comparison-Form, Subjectivity-Type, and Fine-grained Question Subjectivity classification (FQSC). The intent of this comparative evaluation is to spotlight the incremental improvements and the potential enhancements our model contributes to the domain of FQSC. As indicated by the results presented in Table 13, it is important to note that the proposed model consistently outperforms the Yu et al. [7] model across all tested classification paradigms.

**4.3.2. Cross-dataset performance evaluation of RoBERTa, BERT, and XLNet in fine-grained subjectivity tasks.** In this section, the effectiveness of prominent Transformer architectures in the specialized task of FQSC was meticulously evaluated. This comparative analysis included RoBERTa, BERT, and XLNet, which were assessed on Precision, Recall, and F-score metrics. These metrics, averaged over five independent runs, were derived using a 5-fold stratified cross-validation technique to confirm the robustness and generalizability of the evaluation.

Tables 14–16 provide a granular view of the performance of Transformer models RoBERTa, BERT, and XLNet across different tasks and datasets. Table 13 focuses on the FQSC task within the FQSD dataset, showcasing the models' precision, recall, and F-score over five independent runs. Table 14 evaluates these models on the subjectivity-comparison form classification task in the ConvEx-DS dataset, again over five runs, highlighting their robustness and consistency. Lastly, Table 15 examines the models' performance on the subjectivity

**Table 13. Evaluation of the proposed model (Trained on FQSD and tested on Yu et al. [7] dataset) vs. Yu et al. [7] model.**

| Model Type | Model | Precision | Recall | F-Score |
|---|---|---|---|---|
| Subjectivity Classification | Yu et al. [7] model | 75.5 | 61.8 | **68.0** |
| | Proposed model | 92.07 | 90.45 | **90.70** |
| Comparison-Form Classification | Yu et al. [7] model | 91.0 | 90.3 | **90.5** |
| | Proposed model | 98.65 | 98.63 | **98.64** |
| Subjectivity-Type Classification | Yu et al. [7] model | 90.0 | 77.5 | **78.3** |
| | Proposed model | 88.57 | 85.00 | **84.85** |
| Fine-grained Question Subjectivity Classification (FQSC) | - | - | - | - |
| | Proposed model | 88.37 | 84.54 | **85.75** |

**Table 14. Comparative analysis of transformer models' performance (LR = 1e-5) on the FQSC task across the FQSD dataset over five independent runs.**

| Model | K-fold | Precision | Recall | F-Score |
|---|---|---|---|---|
| Roberta | 1 | 96.8 | 96.6 | 96.6 |
| | 2 | 96.0 | 95.9 | 95.8 |
| | 3 | 96.9 | 96.8 | 96.8 |
| | 4 | 97.7 | 97.7 | 97.6 |
| | 5 | 97.8 | 97.8 | 97.8 |
| | Average | **97.04** | **97.16** | **97.12** |
| Bert | 1 | 96.7 | 96.6 | 96.6 |
| | 2 | 95.4 | 95.4 | 95.4 |
| | 3 | 97.5 | 97.5 | 97.5 |
| | 4 | 96.4 | 96.3 | 96.2 |
| | 5 | 96.5 | 96.5 | 96.5 |
| | Average | **96.50** | **96.46** | **96.44** |
| XLnet | 1 | 97.1 | 97.0 | 97.0 |
| | 2 | 97.0 | 97.1 | 97.0 |
| | 3 | 97.4 | 97.4 | 97.4 |
| | 4 | 96.2 | 96.2 | 96.2 |
| | 5 | 97.0 | 96.8 | 96.9 |
| | Average | **96.94** | **96.91** | **96.90** |

classification task in the SubjQA dataset, detailing domain-specific precision, recall, and F-scores to underline their adaptability and effectiveness across diverse contexts.

In Tables 16 and 17, experiments conducted on the SubjQA Dataset were performed with a learning rate (lr) of 3e-5, in contrast to the learning rate of 1e-5 used in other tables, notably

**Table 15. Comparative analysis of transformer models' performance (LR = 1e-5) on the subjectivity-comparison form classification task across the ConvEx-DS dataset [9] over five independent runs.**

| Model | K-fold | Precision | Recall | F-Score |
|---|---|---|---|---|
| Roberta | 1 | 89.23 | 90.10 | 89.66 |
| | 2 | 91.42 | 89.25 | 90.32 |
| | 3 | 90.33 | 89.37 | 89.85 |
| | 4 | 90.44 | 91.26 | 90.84 |
| | 5 | 91.18 | 90.21 | 90.69 |
| | Average | **90.52** | **90.04** | **90.07** |
| Bert | 1 | 83.02 | 81.76 | 81.85 |
| | 2 | 83.15 | 83.64 | 83.09 |
| | 3 | 84.46 | 84.59 | 84.44 |
| | 4 | 79.25 | 80.18 | 78.75 |
| | 5 | 85.45 | 84.54 | 84.90 |
| | Average | **83.86** | **82.94** | **82.60** |
| XLnet | 1 | 80.92 | 81.44 | 80.54 |
| | 2 | 82.90 | 82.07 | 80.30 |
| | 3 | 81.00 | 81.44 | 79.18 |
| | 4 | 78.64 | 78.61 | 77.90 |
| | 5 | 84.86 | 85.80 | 85.11 |
| | Average | **81.864** | **81.872** | **80.206** |

**Table 16. Comparative analysis of transformer models' performance (LR = 3e-5) on the subjectivity classification task across the SubjQA dataset [8] over five independent runs.**

| Model | Domain | Precision | Recall | F-Score |
|-------|--------|-----------|--------|---------|
| Roberta | Books | 89.69 | 89.95 | 89.79 |
| | Electronic | 92.46 | 92.92 | 92.54 |
| | Grocery | 88.39 | 86.51 | 87.09 |
| | Movies | 94.65 | 94.70 | 94.66 |
| | TripAdvisor | 93.36 | 93.15 | 93.23 |
| | Restaurants | 92.48 | 92.02 | 92.20 |
| | Average | **91.27** | **91.90** | **91.98** |
| XLnet | Books | 86.26 | 86.12 | 86.19 |
| | Electronic | 89.64 | 90.70 | 89.69 |
| | Grocery | 79.87 | 79.53 | 79.69 |
| | Movies | 82.96 | 82.01 | 78.51 |
| | TripAdvisor | 89.14 | 88.58 | 88.80 |
| | Restaurants | 93.79 | 89.57 | 90.55 |
| | Average | **86.27** | **86.63** | **86.23** |
| Bert | Books | 90.44 | 87.08 | 87.96 |
| | Electronic | 92.01 | 89.38 | 90.24 |
| | Grocery | 85.06 | 85.58 | 85.26 |
| | Movies | 92.71 | 92.59 | 92.26 |
| | TripAdvisor | 91.06 | 91.32 | 91.12 |
| | Restaurants | 92.93 | 91.41 | 91.87 |
| | Average | **90.99** | **89.60** | **89.51** |

Table 11. This deviation was strategically chosen due to the unique characteristics of the SubjQA dataset, where BERT and XLNet exhibited tendencies to overfit. To ensure a fair comparison across models, the same lr of 3e-5 was adopted for RoBERTa when evaluating this particular dataset, despite RoBERTa not showing the same overfitting issues. This adjustment underscores our commitment to maintaining equitable experimental conditions and accurately assessing model performance across diverse datasets.

Table 17 clearly outlines the performance metrics for each model across a variety of datasets, namely SubjQA, ConvEx-DS, and FQSD. The tasks range from general subjectivity classification to the more nuanced subjectivity-comparison form classification. The learning rate (lr) for each model is noted, revealing the tailored optimization strategies for each dataset. In

**Table 17. Comparative analysis of transformer models' performance on fine-grained subjectivity tasks across multiple datasets over five independent runs.**

| Domain | Task | LR | Model | Precision | Recall | F-Score |
|--------|------|-----|-------|-----------|--------|---------|
| SubjQA [8] | Subjectivity Classification | 3e-5 | Roberta | 91.27 | 91.90 | **91.98** |
| | | | XLnet | 86.27 | 86.63 | **86.23** |
| | | | Bert | 90.99 | 89.60 | **89.51** |
| ConvEx-DS [9] | Subjectivity-ComparisonForm Classification | 1e-5 | Roberta | 90.52 | 90.04 | **90.07** |
| | | | XLnet | 81.86 | 81.87 | **80.21** |
| | | | Bert | 83.86 | 82.94 | **82.60** |
| FQSD | Fine-grained Question Subjectivity Classification (FQSC) | 1e-5 | Roberta | 97.04 | 97.16 | **97.12** |
| | | | XLnet | 96.94 | 96.91 | **96.90** |
| | | | Bert | 96.50 | 96.46 | **96.44** |

light of the varying performances of BERT and XLNet—each outperforming the other in different contexts—it's RoBERTa that consistently demonstrates superiority and steadiness across all evaluated tasks and datasets. This observation highlights RoBERTa's robustness and its capability to maintain high performance levels, making it a standout model for fine-grained question subjectivity classification and related tasks.

RoBERTa's architectural sophistication and comprehensive training regime are central to its preeminence. The dynamic masking feature is a significant enhancement over BERT's static masking and XLNet's permutation-based learning, providing RoBERTa with unmatched contextual flexibility. This innovation is pivotal in enabling RoBERTa to capture the fine distinctions of subjectivity within questions across all datasets examined.

The focused approach of RoBERTa on the masked language model task, to the exclusion of other tasks like NSP, directly correlates with its elevated performance levels. The extensive and varied corpus on which RoBERTa was trained has endowed the model with an intricate understanding of language, evident in its exceptional performance on datasets like FQSD.

In conclusion, this thorough analysis reasserts RoBERTa's role as the definitive model for FQSC. Its consistent top-tier performance, advanced architectural features, and targeted training approach highlight its unmatched ability to discern nuanced subjectivity within questions, evidencing its practical utility and adaptability to diverse datasets.

## 5. Conclusion and future work

In conclusion, the research addresses critical gaps in the area of FQSC. The FQSD, an extensive dataset comprising 10,000 questions, was introduced to serve as a benchmark for future work in this domain. The multi-layered labeling scheme offers depth by categorizing questions not only as subjective or objective but also by their Subjective-types and Comparison-forms. The dataset underwent rigorous validation by a team of three annotators and demonstrated robust inter-annotator reliability, as evidenced by a Fleiss's Kappa score of 0.76 and Pearson correlation values up to 0.80.

FQSD was compared against existing datasets, and it was found to excel in terms of scale, linguistic diversity, and syntactic complexity. Additionally, the visual methodologies offer a nuanced understanding of the dataset and its categories. Experiments with transformer-based models like BERT, XLNET, and RoBERTa further underscored the dataset's effectiveness, with RoBERTa achieving an F1-score of 97%. Furthermore, utilizing LIME provided insights into the models' decision-making, enhancing trust in the findings and experimental outcomes.

This research serves as an initial step towards the ultimate goal of creating high-quality ASQA systems and opens new avenues in various areas where fine-grained subjectivity analysis is crucial. Specifically, it could be employed in tasks such as entity extraction, aspect extraction, comparative relation extraction, and comparative preference classification, among others. These applications could further refine question subjectivity classifications and deepen our understanding of the nuances involved.

## Supporting information

**S1 File.**
(ZIP)

## Author Contributions

**Conceptualization:** Marzieh Babaali, Afsaneh Fatemi, Mohammad Ali Nematbakhsh.

**Data curation:** Marzieh Babaali, Afsaneh Fatemi, Mohammad Ali Nematbakhsh.

**Formal analysis:** Marzieh Babaali, Afsaneh Fatemi.

**Investigation:** Marzieh Babaali, Afsaneh Fatemi.

**Methodology:** Marzieh Babaali, Afsaneh Fatemi.

**Project administration:** Afsaneh Fatemi, Mohammad Ali Nematbakhsh.

**Resources:** Marzieh Babaali.

**Software:** Marzieh Babaali.

**Supervision:** Afsaneh Fatemi, Mohammad Ali Nematbakhsh.

**Validation:** Marzieh Babaali, Afsaneh Fatemi, Mohammad Ali Nematbakhsh.

**Visualization:** Marzieh Babaali.

**Writing – original draft:** Marzieh Babaali.

**Writing – review & editing:** Marzieh Babaali, Afsaneh Fatemi, Mohammad Ali Nematbakhsh.

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
