## [Decision Letter · Decision Letter 0]

11 Jan 2024

PONE-D-23-33628Creating and Validating the Fine-Grained Question Subjectivity Dataset (FSQD): a new benchmark for enhanced Automatic Subjective Question Answering systemsPLOS ONE

Dear Dr. Fatemi,

Thank you for submitting your manuscript to PLOS ONE. After careful consideration, we feel that it has merit but does not fully meet PLOS ONE’s publication criteria as it currently stands. Therefore, we invite you to submit a revised version of the manuscript that addresses the points raised during the review process.

We appreciate the effort and expertise you have brought to your work; however, several areas have been identified that need significant improvement. Please find the detailed and actionable comments below. The comments from point three onwards are of particular importance. These suggestions are aimed at enhancing the clarity, depth, and overall quality of your manuscript. I concur with the reviewer comments and believe that addressing these points will substantially strengthen your paper. 

We look forward to receiving your revised manuscript.

Kind regards,

John Blake, PhD

Academic Editor

PLOS ONE

3. In your Methods section, please include additional information about your dataset and ensure that you have included a statement specifying whether the collection and analysis method complied with the terms and conditions for the source of the data.

4. Please note that PLOS ONE has specific guidelines on code sharing for submissions in which author-generated code underpins the findings in the manuscript. In these cases, all author-generated code must be made available without restrictions upon publication of the work. Please review our guidelines at https://journals.plos.org/plosone/s/materials-and-software-sharing#loc-sharing-code and ensure that your code is shared in a way that follows best practice and facilitates reproducibility and reuse.

5. We note that your Data Availability Statement is currently as follows: [All relevant data are within the manuscript and its Supporting Information files.]

6. We note that Figures 3, 4, 5 and 6 in your submission contain copyrighted images. All PLOS content is published under the Creative Commons Attribution License (CC BY 4.0), which means that the manuscript, images, and Supporting Information files will be freely available online, and any third party is permitted to access, download, copy, distribute, and use these materials in any way, even commercially, with proper attribution. For more information, see our copyright guidelines: http://journals.plos.org/plosone/s/licenses-and-copyright.

1. You may seek permission from the original copyright holder of Figures 3, 4, 5 and 6 to publish the content specifically under the CC BY 4.0 license.

Reviewers' comments:

Reviewer's Responses to Questions

**Comments to the Author**

1. Is the manuscript technically sound, and do the data support the conclusions?

Reviewer #1: Yes

2. Has the statistical analysis been performed appropriately and rigorously? 

Reviewer #1: N/A

3. Have the authors made all data underlying the findings in their manuscript fully available?

Reviewer #1: Yes

4. Is the manuscript presented in an intelligible fashion and written in standard English?

Reviewer #1: Yes

5. Review Comments to the Author

Reviewer #1: The paper under discussion is indeed quite intriguing and contributes significantly to the field of Question Answering (QA) Natural Language Processing (NLP). However, there are some areas where the authors can make improvements to enhance the overall quality and clarity of their work:

1. Visualization Choice and Accessibility:

The usage of word clouds in scientific works is not very common nowadays. It is advisable to replace them with more informative and visually accessible alternatives. Options such as Term Frequency-Inverse Document Frequency (TF-IDF) representations, t-Distributed Stochastic Neighbor Embedding (t-SNE), or topic modeling could provide richer insights. Moreover, it's important to ensure that the chosen visualizations are color-blind friendly to make the paper accessible to a wider audience. Lastly, it's essential that the visualizations effectively convey the intended message.

2. Figure Quality and Explanations:

The figures in the paper need to be improved. They lack informative captions, making it challenging for readers to understand their content. It's crucial to provide clear and concise explanations for each figure, including details on how certain metrics, like lexical count in Figure 8 or the measurement of syntactic and semantic dimensions, were derived. Authors should briefly describe the techniques or methodologies employed to calculate these measures, making it easier for readers to replicate or understand the experiments.

3. Dataset Schema:

Including the schema of the dataset in a JSON or similar format would be beneficial. This would provide readers with a clearer understanding of the data used in the experiments and help them better interpret the results.

4. Explanation of LIME Evaluations:

The paper should offer an intuitive explanation of LIME (Local Interpretable Model-agnostic Explanations) evaluations. This will help readers comprehend the significance and implications of using LIME in the context of QA NLP work. Additionally, it's advisable to reference related works like https://doi.org/10.1186/s12859-022-04751-6

in the literature that discuss the use of QA and NLP evaluation.

5. Benchmark Choice:

The authors should provide a clear rationale for choosing RoBERTa as the benchmark model. Explaining why RoBERTa was selected over other models can help readers understand the motivations behind this choice and the implications for the study. Additionally, it would be valuable to mention whether the QA systems were evaluated using exact match metrics or similarity metrics based on BERT or other models.

6. Data Quality Section:

While the paper briefly touches on data quality concerns, it would be beneficial to have a dedicated section that thoroughly addresses data quality issues. This can include discussions any potential biases in the dataset or annotators biases and how it is dealth. Providing this information will enhance the transparency and credibility of the research.

Please proof read the paper again and bring quality figures with captions and how these are made (used some techniques like for visual?)

6. PLOS authors have the option to publish the peer review history of their article (what does this mean?). If published, this will include your full peer review and any attached files.

Reviewer #1: **Yes: **Shaina Raza

---

## [Author Response · Author response to Decision Letter 0]

29 Feb 2024

Dear Editor,

Thank you for considering our manuscript and for providing us with the opportunity to revise and resubmit it to PLOS ONE. We are grateful for the valuable feedback received from you and the reviewers. We believe that addressing these comments will greatly enhance the clarity, depth, and overall quality of our work.

We have carefully reviewed each point raised and have undertaken significant revisions to our manuscript to address the concerns highlighted. In particular, we have paid close attention to the comments from point three onwards, as per your recommendation. We have addressed each specific comment made by the reviewers and editor in our detailed 'Response to Reviewers' document, ensuring that our manuscript meets the high standards of PLOS ONE.

In response to the editorial feedback, we have included a comprehensive legend for Figure 1, providing a clear and concise description of its contents and significance in relation to our study. This addition aims to enhance the figure's understanding and ensure its alignment with the manuscript's narrative.

In accordance with the submission guidelines, upon resubmitting our revised manuscript by the specified deadline, we will provide the following items:

• Rebuttal Letter: A detailed letter responding to each point raised by the academic editor and reviewers. This document will be uploaded as a separate file labeled 'Response to Reviewers'.

• Marked-Up Manuscript: A copy of our manuscript with track changes, highlighting the revisions made. This document will be uploaded as a separate file labeled 'Revised Manuscript with Track Changes'.

• Unmarked Manuscript: An unmarked version of the revised paper without tracked changes. This document will be uploaded as a separate file labeled 'Manuscript'.

We are committed to enhancing the reproducibility and transparency of our research and appreciate the guidance provided by PLOS ONE in this process.

Thank you for bringing to our attention the potential citation advantage associated with depositing data in a repository, as highlighted in your recent publication (https://doi.org/10.1371/journal.pone.0230416). We greatly appreciate this valuable insight and the opportunity to enhance the impact and accessibility of our research.

Data Repository Deposition:

We are pleased to announce the deposition of our raw data in a GitHub repository, adhering to PLOS's open data policies. This initiative underscores our commitment to open science by ensuring our data is accessible to the research community. The dataset can be found at https://github.com/mahsamb/FSQD, and the code related to our figures and model is at: https://github.com/mahsamb/FSQD/blob/main/Figures.ipynb.

We look forward to the opportunity to contribute to the PLOS ONE community and thank you again for your valuable feedback. Please do not hesitate to contact us if further information or clarification is required in the meantime.

Kind regards,

Dr. Afsaneh Fatemi

---

## [Decision Letter · Decision Letter 1]

20 Mar 2024

Creating and Validating the Fine-Grained Question Subjectivity Dataset (FSQD): a new benchmark for enhanced Automatic Subjective Question Answering systems

PONE-D-23-33628R1

Dear Dr. Fatemi,

We’re pleased to inform you that your manuscript has been judged scientifically suitable for publication and will be formally accepted for publication once it meets all outstanding technical requirements.

Kind regards,

John Blake, PhD

Academic Editor

PLOS ONE

Reviewers' comments:

Reviewer's Responses to Questions

**Comments to the Author**

1. If the authors have adequately addressed your comments raised in a previous round of review and you feel that this manuscript is now acceptable for publication, you may indicate that here to bypass the “Comments to the Author” section, enter your conflict of interest statement in the “Confidential to Editor” section, and submit your "Accept" recommendation.

Reviewer #1: All comments have been addressed

2. Is the manuscript technically sound, and do the data support the conclusions?

Reviewer #1: Partly

3. Has the statistical analysis been performed appropriately and rigorously? 

Reviewer #1: N/A

4. Have the authors made all data underlying the findings in their manuscript fully available?

Reviewer #1: Yes

5. Is the manuscript presented in an intelligible fashion and written in standard English?

Reviewer #1: Yes

6. Review Comments to the Author

Reviewer #1: Most comments are addressed

Figures can be enchanced.

Details for data should be mentioned .

Authors should highlight the contribution in abstract too

7. PLOS authors have the option to publish the peer review history of their article (what does this mean?). If published, this will include your full peer review and any attached files.

Reviewer #1: No

---

## [Editor Report · Acceptance letter]

2 May 2024

PONE-D-23-33628R1 

PLOS ONE

Dear Dr. Fatemi, 

I'm pleased to inform you that your manuscript has been deemed suitable for publication in PLOS ONE. Congratulations! Your manuscript is now being handed over to our production team.

Kind regards, 

on behalf of

Dr. John Blake 

Academic Editor

PLOS ONE